# A bacterial ecocline in *Klebsiella pneumoniae* may explain its backboned phylogeny

**Siqi Liu, Sarah L. Svensson, Daniel Falush** ⓘ*

Shanghai Institute of Materia Medica, Chinese Academy of Sciences, Shanghai, China

* daniel.falush@ips.ac.cn

## Abstract

The genetic structure of bacterial species is most often interpreted in terms of demographic processes such as clonal descent, but can also reflect natural selection and hence give functional and ecological insight. *Klebsiella pneumoniae* (KP) disperses effectively around the world and has high recombination rates, which should result in the species having a well-mixed gene pool. Nevertheless, phylogenies based on diverse KP strains contain a "backbone." This structure reflects a component of variation where the first component in Principal Components Analysis (PCA), PC1, explains 16.8% of the total variation. We propose that the component reflects a "bacterial ecocline" generated by diversifying selection on a quantitative genetic trait. We simulated the evolution of a bacterial population with a polygenic quantitative trait, where strains with the most extreme trait values have a small advantage. These simulations can recapitulate our KP PCA results and other features of its genetic diversity. As well as providing an explanation for the phylogenetic backbone, our results provide insight into how species such as KP can speciate, via stronger selection on the trait or a reduction in gene flow. Our hypothesis that there is a bacterial ecocline in KP raises two questions, namely what the trait is underlying it and why is the trait under diversifying selection? The genes that are most strongly associated with PC1 provide some hints, with the top locus encoding Kpa fimbriae. Identification of the trait, if it exists, should facilitate insight into selection on quantitative genetic traits in natural bacterial populations, which have largely been unstudied in microbiology, except in the atypical context of antibiotic resistance.

## Introduction

What information can be gleaned from phylogenetic trees [1]? The shape of a tree reflects key demographic factors, such as clonal relationships or geographic structure shaped by barriers to dispersal and genetic exchange [2]. Deep splits within trees and an absence of intermediate strains can, for example, reflect long-standing divergence between groups that has arisen due to niche specialization and/or low levels of

**Data availability statement:** All scripts and data can be found at GitHub (https://github.com/sql647/KpBACKBONE) and are archived at Zenodo (https://doi.org/10.5281/zenodo.18520201).

**Funding:** This work was supported by the National Natural Science Foundation of China (NSFC; grant 32350710791 to D.F.). The funders had no role in study design, data collection and analysis, decision to publish, or preparation of the manuscript.

**Competing interests:** The authors have declared that no competing interests exist.

**Abbreviations:** COG, Clusters of Orthologous Groups; HGT, Horizontal Gene Transfer; KP, *Klebsiella pneumonia*; LD, Linkage Disequilibrium; MAF, minimum allele frequency; MIC, minimum inhibitory concentration; PCA, Principal Components Analysis; VP, *Vibrio parahaemolyticus*.

genetic exchange [3,4]. However, many species show a pattern of differentiation that is more continuous in character, which can be harder to explain.

*Klebsiella pneumoniae* (KP) is a major human pathogen capable of causing severe infections in immunocompromised individuals [5], but is found in a wide variety of environments [6], including soil [7], wastewater [8], and even the ocean [9]. Like many other bacterial species, KP has an enormous population size, exhibits high transmission efficiency, and has high recombination rates between strains [10]. As a result, its global genetic diversity can be captured from samples collected in a single region, whether from Europe [11], Africa [12], Asia [13], or elsewhere. Despite these features, as we will describe in detail below, the phylogenetic tree of a broad sample of KP strains shows deep structure, with strains radiating from multiple points along a "backbone." Until now, to our knowledge, no explanations have been proposed for this shape.

We propose that the shape of the phylogenetic tree of KP reflects diversifying selection [14] acting on a quantitative trait [15]. Specifically, we simulate a population of bacteria with a single phenotypic trait. Genetic variants influence the trait additively, meaning that specific variants increase or decrease trait values by a fixed amount that does not depend on the presence or absence of other variants. We show that if strains exhibiting extreme trait values have a fitness advantage, we can recapitulate patterns of diversity found in KP. We call the gradient in variant frequency values that arises a "bacterial ecocline." Quantitative traits are central to our understanding of genetic diversity in animals and plants, but have hitherto been largely overlooked for bacteria. If the ecocline hypothesis is correct and we can identify the trait or traits under selection in KP, it should provide considerable insight into the genetic architecture of bacterial quantitative traits as well as the fitness trade-offs [16] that the species experiences.

## Results

### The phylogenetic backbone reflects a component of variation

Genetic relationships amongst bacteria are most often represented using a phylogenetic tree [17]. However, the interpretation of these trees is complicated by the presence of high levels of homologous recombination in many species, which violates the assumption of vertical descent made by phylogenetic methods. Consequently, trees are best thought of as heuristic tools to visually summarize the pairwise genetic distances between strains and the internal nodes, especially deeper ones, should *not* be thought of as representing putative ancestral sequences. For example, a core genome SNP–based phylogenetic tree from a global collection of KP isolates exhibits a "star-like" topology, with many branches of approximately equal length (Fig 1A, non-redundant dataset with clonally related strains removed). SNP distance heatmaps and fineSTRUCTURE [18] analyses depict the relationships among isolates in the same non-redundant dataset used for tree inference (S1 Fig), while consistent backbone topologies are recovered using different phylogenetic reconstruction algorithms, including IQ-TREE [19] for the non-redundant dataset and FastTree [20]

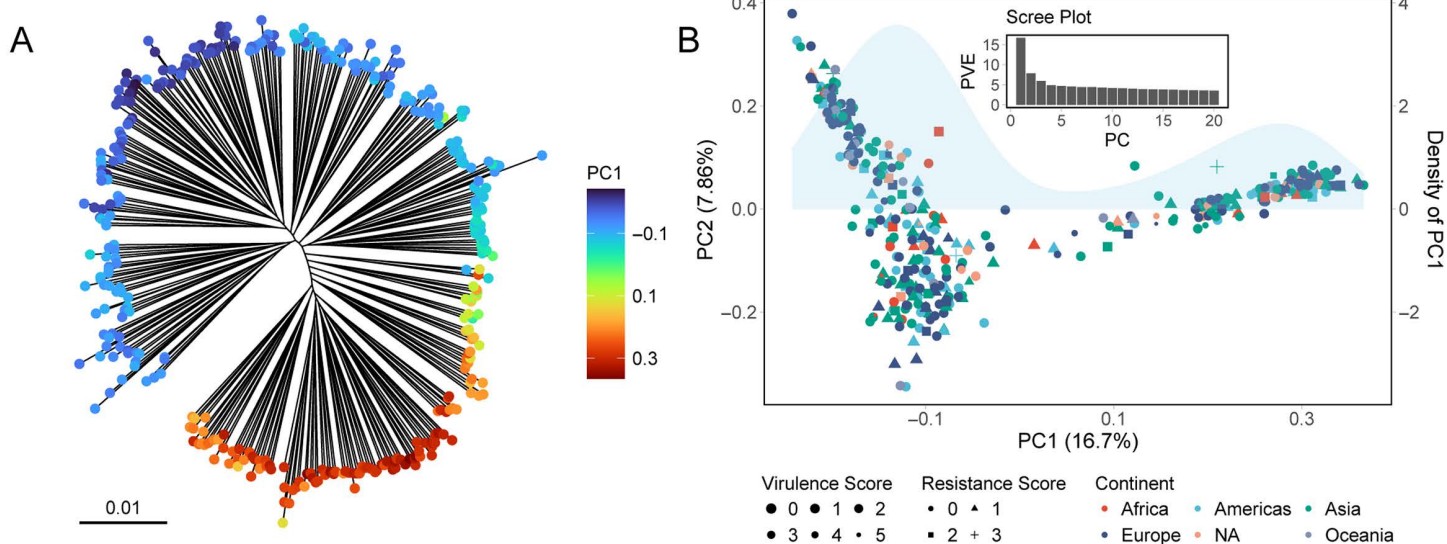

**Fig 1. Population structure and genetic variation in a non-redundant KP dataset.** (A) Maximum-likelihood phylogenetic tree of the non-redundant KP dataset based on core SNPs, with a color gradient showing PC1 variation. **(B)** PCA of the non-redundant KP dataset based on Linkage Disequilibrium (LD) pruned SNPs (see Methods). Colors indicate geographic origin, while marker size and shape represent resistance and virulence scores, respectively, determined by Kleborate [24]. The sky-blue shading represents the density of PC1 values. The inset scree plot shows the variance explained (PVE) by the first 20 PCs.

for the entire dataset (S2 Fig). This "bushy" property of the tree reflects extensive recombination, with the length of the branches reflecting the overall level of diversity in the gene pool [10,21].

Notably, although the KP tree is bushy, the overall shape of the KP tree is oval rather than approximately circular as, for example, seen in *Vibrio parahaemolyticus* (VP) [22,23] and KP lineages, instead of radiating from a single point, are connected by a "backbone" (Figs 1A and S2). A tree has a backbone if there is a single connected path of branches which together makes up a large fraction of the overall internal branch length of the tree. A backbone of this sort is present in any phylogeny drawn based on a diverse sample of KP genomes (Figs 1A and S2), but it may be less visually evident than in Fig 1A, depending on the strains sampled, the rooting of the tree, and the phylogenetic and plotting algorithms.

In the KP tree shown in Fig 1A, the backbone consists of around a dozen branches of similar length. In principle, strains that branch at the top of the tree could be given maximum backbone scores and those at the bottom minimum ones, with intermediate strains given values according to how far along the line they branch. The dozen-or-so backbone scores defined in this way would be roughly equally spaced, approximating a single continuously varying trait. We have not defined scores in this way, instead using Principal Components Analysis (PCA) [25,26], which implicitly assumes a continuous model of variation underlying each component. The data underlying this Figure can be found in https://zenodo.org/records/18520201.

PCA identifies axes of variation that together explain the overall diversity within the species, with the first component, PC1, explaining the largest fraction of variation. When PCA was applied to the non-redundant global KP core genome SNP dataset used to construct the tree in Fig 1A, the first component (PC1) explained 16.7% of the variation, with subsequent components each explaining approximately 8% or less (Fig 1B). PC1 captured the genetic variation underlying the phylogenetic backbone of the species, since strains with extreme PC1 values are found at either end of the backbone, while strains branching from the middle have intermediate values (Fig 1A). Strain loadings for PC1 are approximately continuous, albeit with a paucity of intermediate values (Fig 1B).

PCA is sensitive to all signals of relatedness, including clonal structure and can be prone to overfitting, especially where there are orders of magnitude more genetic markers than strains [27]. However, we found that PC1 was reproducible in analyses using non-overlapping parts of the same dataset (Fig 2). Firstly, PC1 strain loadings were highly correlated (Fig 2A and 2B) if components were defined by strains on the right-hand side of the phylogenetic tree or those on the left (as defined in S3 Fig), implying that they are not generated by residual clonal structure, which should result in the most closely related sets of strains grouping together on the same side of the tree. Secondly, PC1 values obtained when performing analysis on one half of the chromosome were highly correlated with those of PC1 calculated using the other half (Fig 2C and 2D), implying that the signal is not due to specific genetic loci. Thirdly, they were also highly correlated when calculated using SNPs with extreme PC1 loadings or loadings that are closer to 0 (Fig 2E–2H), implying that the signal affects genetic variation genome-wide.

To benchmark these results, we applied the same methods to data from a simulated, neutrally-evolving bacterial population (S4A and S4B Fig), which will be discussed in more detail below. In the simulated data, PC1 did not explain as much of the variance as in the KP data (S4C Fig). Furthermore, only one of the three half-matching tests showed a significant correlation for this dataset (S4 and S5 Fig), in contrast to the consistent signal observed in KP. These results indicate that the simplest neutral scenario we explored does not reproduce the key features of the empirical data.

VP also represents a useful benchmark for half-matching tests because its genetic structure is well characterized. The species has geographical population structure [22], which arises because the species lives in coastal waters and there have been barriers to movement of bacteria between oceans, albeit disrupted by recent human activity. VP also has a differentiated genetic cluster first designated as the EG1a ecogroup [23] but renamed the Molassodon ecospecies [28] since it is qualitatively distinct from other identified variation in VP, showing nearly fixed differences from other Typical strains in approximately 40 core genes while remaining undifferentiated across the rest of the genome [23,28]. This pattern is consistent with frequent recombination between ecospecies homogenizing variation in most of the genome, while natural selection has created and maintained differentiation at ecospecies loci.

When three different sorts of half-matching tests were applied to a global non-redundant VP dataset of, all gave reproducible results that reflected the population structure in the dataset (Figs 3A–3F and S6A–S6E). However, when analysis was restricted to the Asian population (VppAsia), the phylogenetic tree is close to being a perfect circle and PC1 instead reflected ecospecies structure (Figs 3G–3L and S6F–S6H). This structure generated weaker correlations in two of the three tests (Figs 3J and S5H), but there was no correlation between low loading and high loading loci, consistent with a single randomly mixing gene pool for most of the genome [23,28]. Thus, the reproducibility of PC1 in all three tests in KP shows that the structure is genome-wide and, in this respect, is more like geographic population structure and less similar to ecospecies structure, which is only present in a fraction of the genome [28,29].

## Loci most strongly associated with PC1 are localized in the genome

Analysis of PC1 loadings and LD patterns allow us to determine whether the backbone signal is driven by a small set of loci or reflects genome-wide processes. In addition to strain loadings (Fig 1B), SNP loadings were computed from the PCA results, indicating which genomic sites contribute most strongly to each component. SNP loadings for PC1 showed four clear peaks (Fig 4A, left). Each peak contained many SNPs within less than 10 kb (S7 Fig). We also calculated loadings for accessory gene presence-absence, based on a Panaroo-generated [30] pangenome of the non-redundant strains, and found several had high loadings (Fig 4A, right).

To investigate potential fitness effects of the loci most strongly correlated with PC1, we chose a cutoff of 300 for squared PC1 loadings and calculated pairwise LD values for all SNPs and genes above this cutoff, which quantify whether allele combinations at each pair loci segregated independently of each other within the KP population. Hierarchical clustering of LD values organized the extreme loading variants into 44 blocks (Fig 4B). Notably, the top four blocks include a

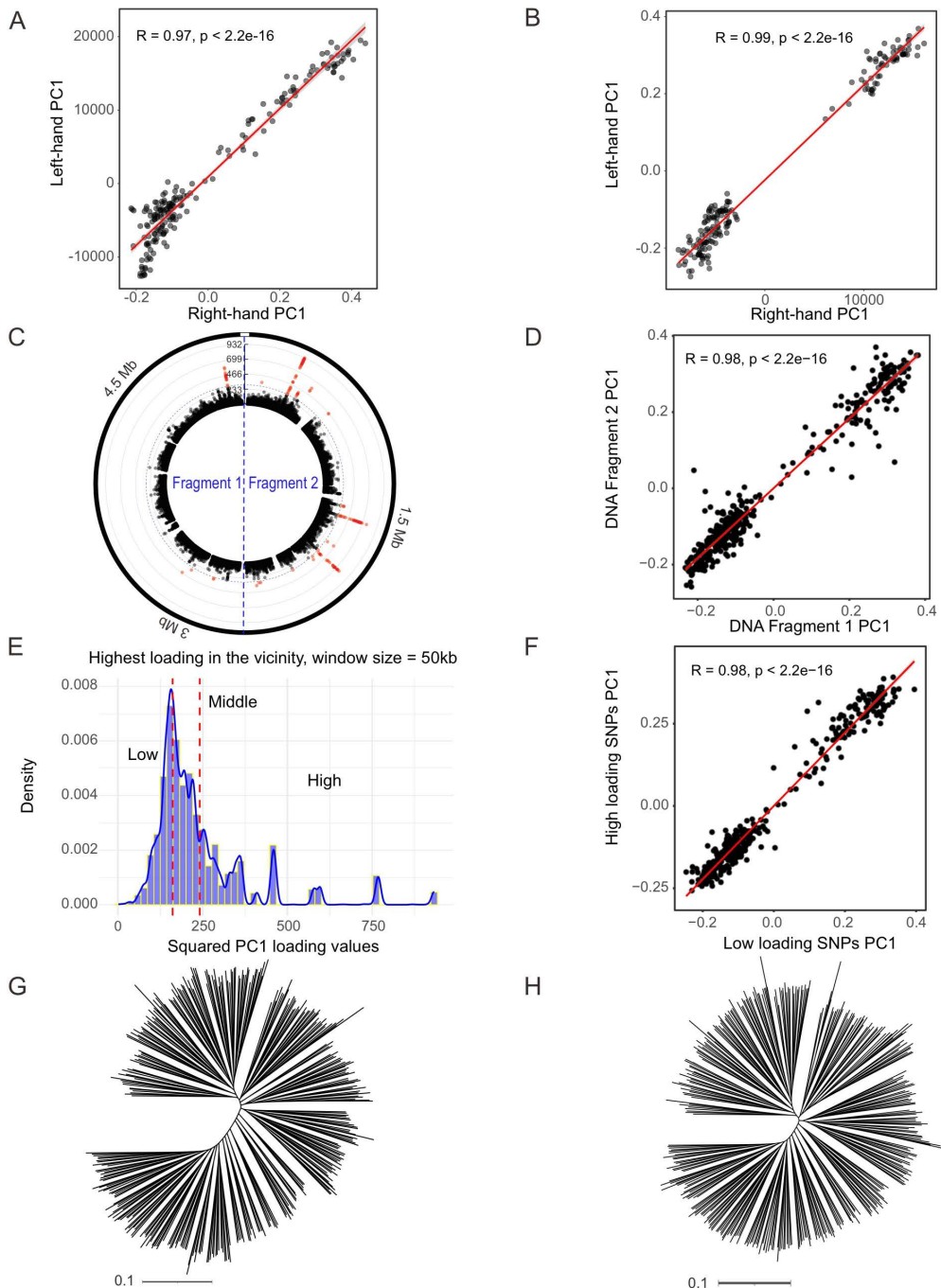

**Fig 2. Three types of half-matching tests for the non-redundant KP dataset. (A, B)** Half-strain matching: (A) Correlation of PC1 from strains on the right hand of the tree projected onto left-hand group results. (B) Correlation of PC1 from the left-hand group mapped onto right-hand group results. **(C, D)** Half-genome Matching: (C) The genome is divided into two fragments: Fragment 1 (1 bp–2.6 Mb) and Fragment 2 (2.6 Mb–5.2 Mb). (D) Correlation of PC1 between the two fragments. **(E–H)** Partial-variants matching: (E) Histogram of SNP loadings on PC1, dividing SNPs into high- and low-loading groups. (F) Correlation of PC1 between high- and low-loading SNP groups. (G) Phylogenetic tree based on low-loading SNPs. (H) Phylogenetic tree based on high-loading SNPs. The data underlying this Figure can be found in https://zenodo.org/records/18520201.

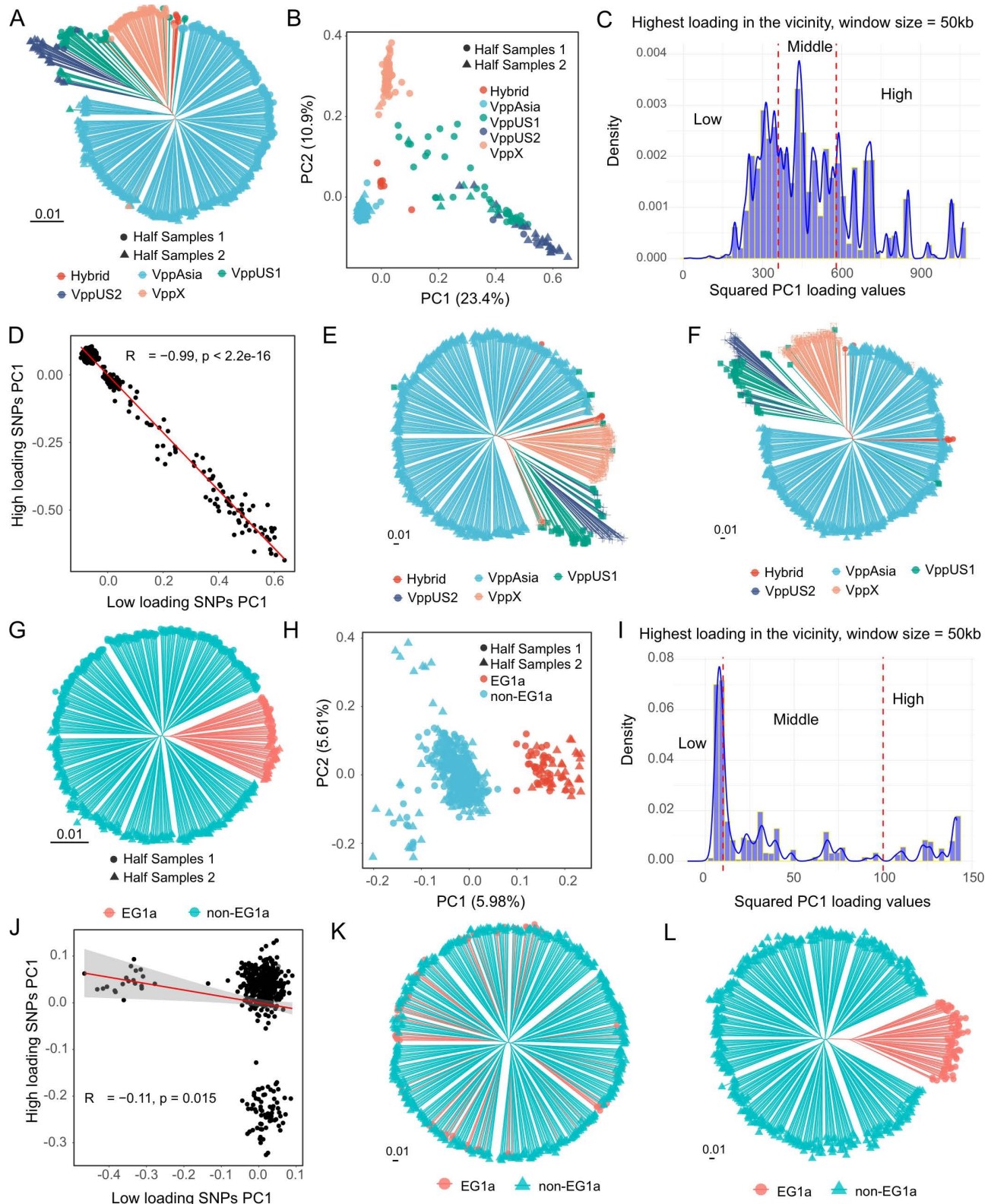

**Fig 3. Partial-variants matching for VP datasets. (A)** Phylogenetic tree of the non-redundant VP dataset. Branch colors indicate populations defined by fineSTRUCTURE [18] (see S6A Fig), and tip nodes represent the half-matching-based dataset partition. The Half-Strain Matching and

Half-Chromosome Matching results are shown in S5 Fig. **(B)** PCA of the non-redundant VP dataset, using the same legend as in panel a. **(C)** Distribution of SNP loadings, with SNPs categorized into low, medium, and high groups based on the maximum loading value within 50 kb. **(D)** Correlation of PC1 between high- and low-loading SNP groups for the non-redundant VP dataset. **(E)** Phylogenetic tree constructed using low-loading SNPs for the non-redundant VP dataset. **(F)** Phylogenetic tree constructed using high-loading SNPs for the non-redundant VP dataset. **(G)** Phylogenetic tree of the VppA-sia dataset. Branch colors indicate ecospecies, and tip nodes represent the half-matching-based dataset partition (see S4 Fig). **(H)** PCA of the dataset of VppAsia, using the same legend as in panel G. **(I)** Distribution of SNP loadings, categorized into low, medium, and high based on loading values. **(J)** Correlation of PC1 between high- and low-loading SNP groups for the VppAsia dataset. **(K)** Phylogenetic tree constructed using low-loading SNPs. **(L)** Phylogenetic tree constructed using high-loading SNPs. The data underlying this Figure can be found in https://zenodo.org/records/18520201.

large fraction of all loci with PC1 values greater than 300, including the highest loading accessory genes. The organization and gene content of these blocks is shown in Figs 4C and S8–S11.

Block 1 (S8 Fig), which has the most extreme loading variants, includes two genes with SNPs (KP1_0337 and KP1_0338). These genes are part of the five-gene *kpa* fimbriae locus, which is part of the core genome in both KP and *K. variicola* [31,32], but found at lower frequency in other *Klebsiella* species. While the Kpa fimbriae are less characterized than those encoded by the *fim* locus, experiments in LM21 (a low PC1-loading strain) found that absence of *kpaC*, which encodes the outer-membrane usher protein required for chaperone-usher fimbrial assembly, significantly reduced in vitro biofilm formation [33]. The Δ*kpaC* mutant was not affected for binding to either a human cell line or *Arabidopsis* seedlings [33]. The most differentiated gene (KP1_0338/*kpaE*) encodes the adhesin subunit. The high and low variants are too distinct to have evolved within KP and cluster separately on a multispecies tree of the locus (S8C Fig), implying that one or both variants was likely imported from another species.

Block 2 (S9 Fig) and block 4 (S10 Fig) also include SNPs in single genes (KP1_1632, encoding a filamentous hemagglutinin N-terminal domain-containing protein and KP1_5387, encoding a winged helix-turn-helix transcriptional regulator, respectively). The former, as for the *Kpa* fimbral adhesin, suggests a difference in adhesion, but so far appears uncharacterized. The regulator encoded by KP1_5387 in block 4 is also unstudied but seems always encoded next to an alcohol dehydrogenase in KP, *K. variicola*, *K. quasipneumoniae*, *K. africana*, and *K. quasivariicola* strains.

Block 3 (S11 Fig) includes both SNPs and accessory genes and can be subdivided into three sub-blocks (3a, 3b, 3c) based on LD patterns. The majority of strains fall into two configurations (S11G Fig), with the high-loading configuration including all accessory gene elements and the low-loading configuration including none, but there are also strains with intermediate configurations.

Block 3 genes are located in two different genomic regions in the reference strain (NTUH-K2044), with block 3b genes encoded at a different location from block 3a and 3c genes (Fig 4C). Blocks 3a and 3c were previously highlighted as associated with invasive strains [34], as well as those implicated in bovine mastitis [35]. Key genes of block 3a are *kfuABC*, encoding a ferric iron ABC transporter, which aids iron acquisition and impacts mouse colonization [34,36]. The block also encodes two c-di-GMP-related proteins, a regulator, and glycogen/PTS (phosphotransferase) sugar utilization/uptake genes. Block 3c is adjacent to block 3a in the genome and contains genes related to sugar uptake, metabolism, and regulation (again via c-di-GMP). Block 3b contains accessory genes encoding a cellulose synthase, three DUF3131 proteins, OpgC (osmoregulated glucan production), and several regulatory proteins (hybrid sensor kinase, anti-sigma-factor, and its antagonist).

## Functional enrichment analysis

To establish whether PC1 exerts a greater influence on core or accessory genome variation, we compared the distribution of absolute PC1 loading values between core SNPs, core genes, and accessory genes. The dashed vertical lines indicate the mean loading value for each group, which are closely aligned, suggesting similar overall contributions to PC1 (Fig 5A). However, accessory genes display a longer right tail in the distribution, indicating the presence of more extreme loading

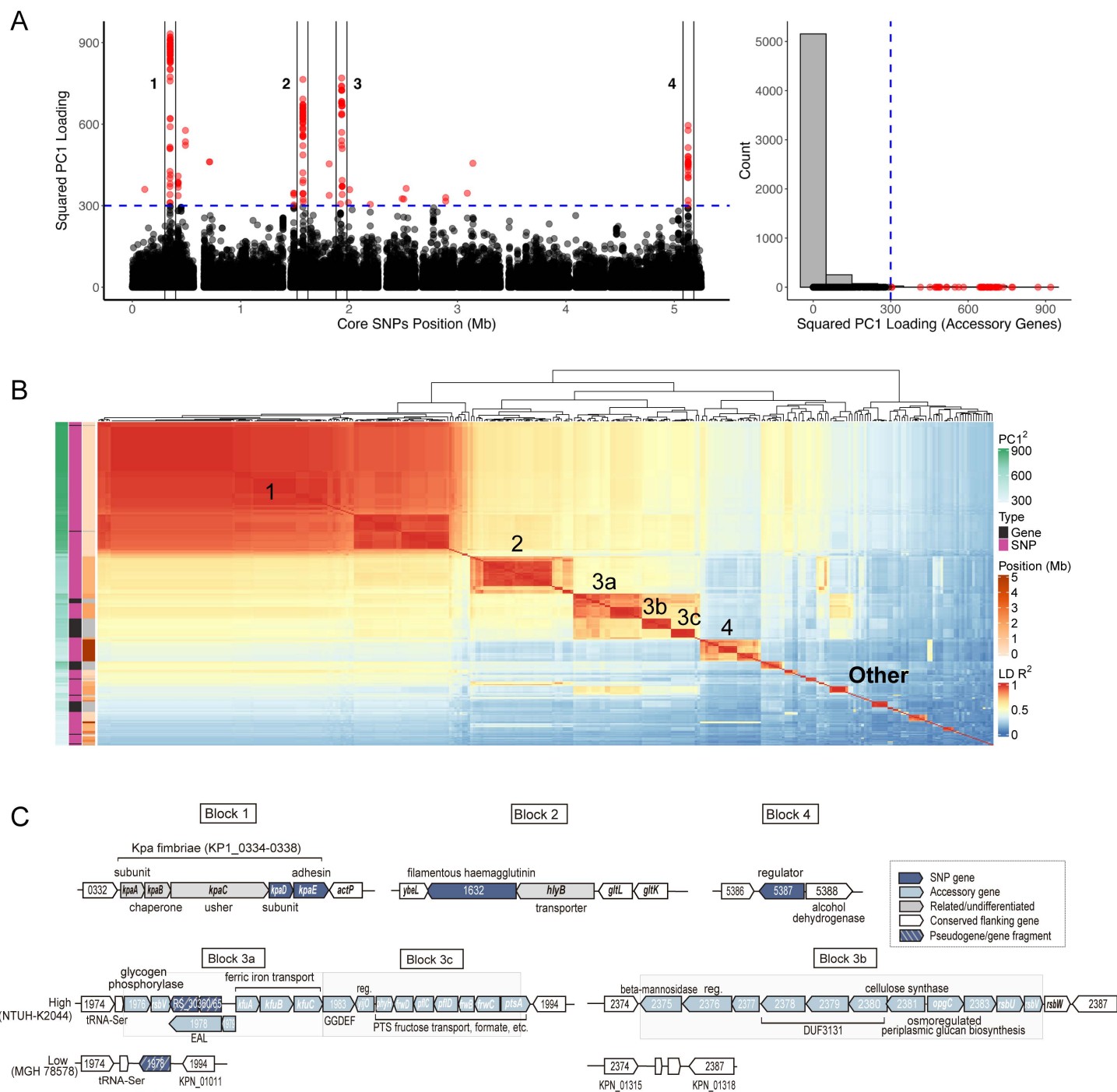

**Fig 4. Location, function, and linkage disequilibrium of PC1-associated loci. (A)** Manhattan plot of squared core SNP (left) and accessory gene (right) loadings for PC1, with the left panel using NTUH-K2044 as the coordinate reference. Variants with values greater than 300 are defined as high-loading variants and are marked in red. **(B)** Heatmap of pairwise LD $R^2$ values for high-loading variants, with the first four blocks (including sub-blocks) labeled. **(C)** Functional annotation of blocks 1–4. The data underlying this Figure can be found in https://zenodo.org/records/18520201.

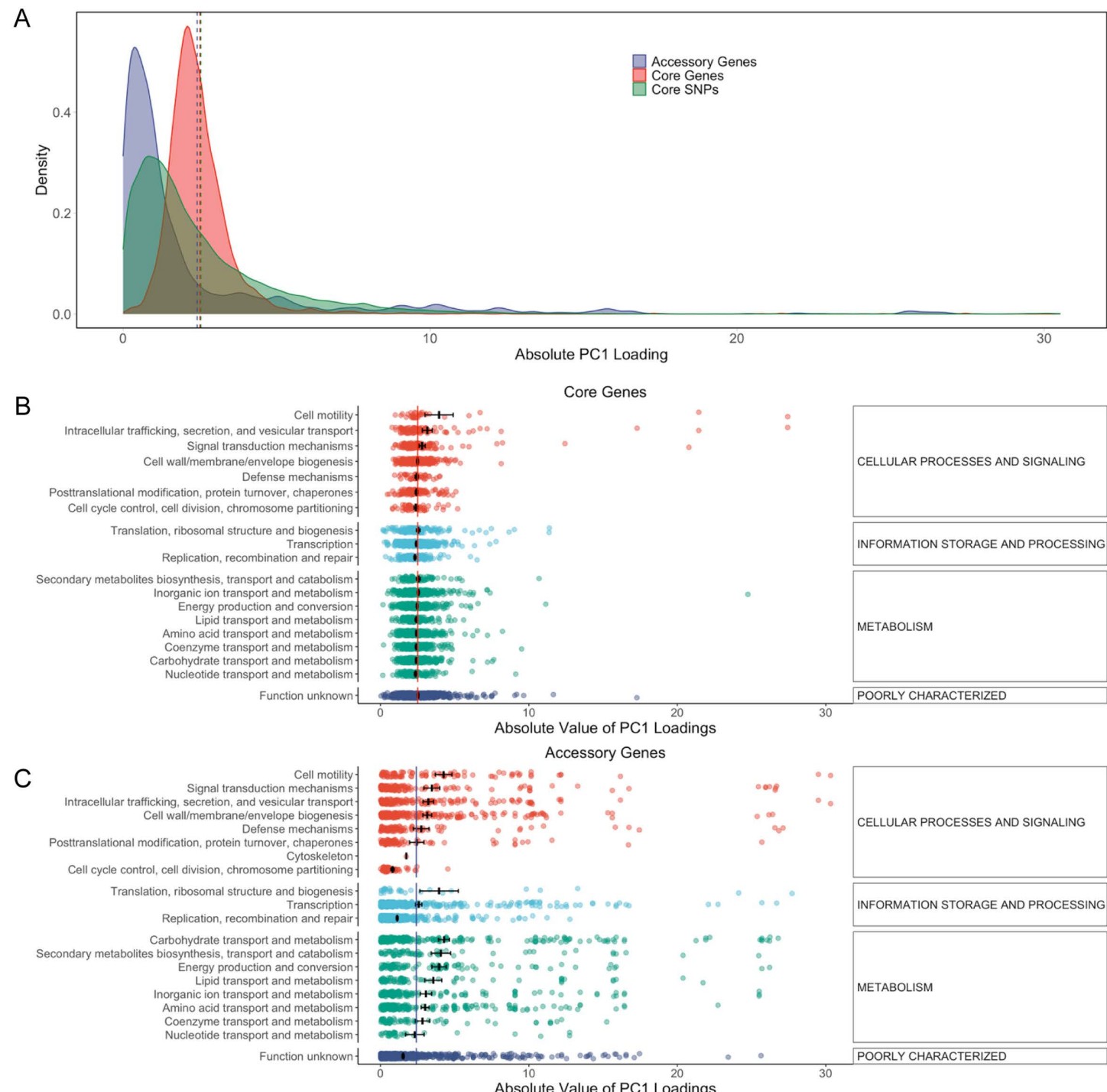

**Fig 5. Overall and COG-specific distributions of absolute PC1 loading values for variants. (A)** Density plot of PC1 loading values for core SNPs, core genes, and accessory genes. Mean absolute PC1 loadings of core SNPs, **(B)** core genes, and **(C)** accessory genes grouped by COG category. Bars represent the average absolute loading within each COG category, with error bars indicating the standard error. The dashed (A) and solid (B) lines mark the overall mean absolute loading across all features (genes or SNPs, respectively). The data underlying this Figure can be found in https://zenodo.org/records/18520201.

values. This suggests that, although the average contribution is similar, a subset of accessory genes is more strongly influenced by PC1.

To establish whether there is a systematic relationship between gene function and SNP loadings, we calculated the mean absolute SNP loading for genes in each Clusters of Orthologous Groups (COG) functional category (Fig 5B) and found almost no difference between categories. However, larger and statistically significant differences were found for accessory genes (Fig 5C), with metabolic genes showing consistently higher mean absolute loading values than those involved in recombination and genome repair or uncharacterized genes.

### Virulence, antibiotic, and isolation source profiles provide few clues

Although virulence and antibiotic resistance are important for the clinical assessment of KP, our analysis suggests that they are not the primary determinants of its overall population structure, which is not unexpected given that most lineages only cause human disease sporadically. KP is a widespread environmental and opportunistic species, commonly found in diverse non-human habitats and in the gut microbiota of healthy individuals, where selective pressures related to clinical infection are limited. Fig 1B shows the isolate metadata and Kleborate [24]-assigned resistance and virulence scores for the non-redundant dataset. Strains do not cluster based on geography, resistance, or virulence.

For multi-host bacteria, host adaptation can be an important evolution [37,38]. KP can be isolated from a variety of sources, including animals, plants, humans (which represent the largest proportion in databases), and water, including the sea. However, there is no strong evidence to support that host selective pressure is responsible for shaping the backbone of the KP phylogenetic tree.

Based on KP1_0338, the gene most strongly associated with PC1, we classified the multi-source strains into high and low variant groups (S12A Fig). A Fisher's exact test revealed a significant association between KP1_0338 variant group and host source ($p < 0.01$), primarily driven by strains from water and horses (S1 Table). However, we found that most water-isolated strains were from Manchester, where they were collected from patient or hospital wastewater sources. According to the Kleborate [24] annotation, most of these strains had a resistance score of 2 and a virulence score of 0 or 1, indicating that they are typical hospital-associated clones. The strains mostly belong to a handful of clonal lineages, and we think the PC1 association is most likely a sampling artifact.

The KP isolates from horses were predominantly collected in France and Iceland, from various anatomical sites through both post-mortem examinations and live-animal sampling. However, when we projected the combined dataset—including the non-redundant dataset, along with strains isolated from water and horse sources—onto the results corresponding to Figs 1B and S12B, the projection revealed that the distribution of horse-derived strains was not significantly different from that of the non-redundant dataset. Thus, the results provide preliminary evidence for an association of the KP1_0338 locus with colonizing horses, but no evidence for a host association of PC1 as a whole.

In summary, our analysis finds no evidence that the observed population structure of KP is shaped by simple host-specific selective pressures. Moreover, no clear associations were observed between population structure and geographic origin, antimicrobial resistance, or virulence. Instead, the distribution of KP1_0338 appears to be largely independent of host origin, indicating that other ecological or evolutionary factors that are not captured by current metadata may underlie the observed patterns.

### The backbone is not reproduced by the simple neutral models we explored

The presence of this substantial component of variation raises the question: how has it arisen? Simple models of a bacterial population evolving with mutation and homologous recombination failed to recapitulate the patterns we observed in KP. For example, simulations with a population size of 2,000 and one homologous recombination event per genome of size 10 kb approximately matched the overall LD decay curve seen in the real data (Figs 6A and S14). However, for these parameters, there was no phylogenetic backbone evident, and PC1 failed to explain as much variance as in the

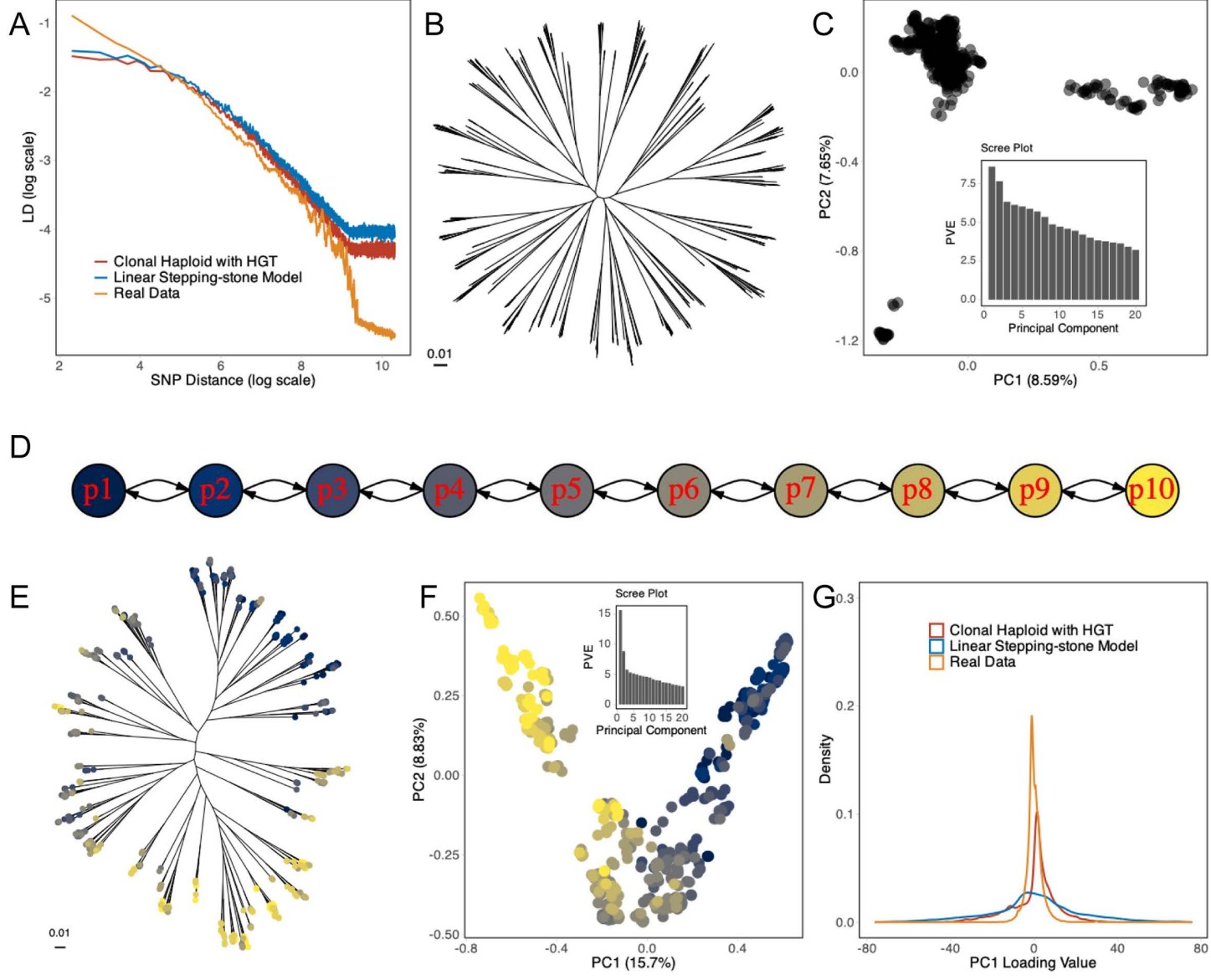

**Fig 6. Simulated neutral models. (A)** LD decay comparison to real data for best-fit neutral single population and 10-population stepping-stone models. Phylogenetic tree **(B)** and PCA results **(C)** for single population evolving neutrally. **(D)** Illustration of the 10-population stepping-stone model. Phylogenetic tree **(E)** and PCA **(F)** for the best fit stepping-stone model, coloured as shown in (D). **(G)** Density plot of SNP PC1 loading values illustrating the distribution of SNP loadings for real data and two neutral models. The data underlying this Figure can be found in https://zenodo.org/records/18520201.

real data (Fig 6B and 6C). As noted above, PC1 was also not reproducible when applied to different halves of the tree in a half-matching test (S4B–S4E Fig). More generally, we did not see a phylogenetic backbone in any simulations of a single neutrally evolving population (representative simulations shown in S13 Fig).

The model of evolution used to generate the data in Fig 6A assumes that recombination rates are uniform across the species. More complex models are possible, for example, in which recombination rates are high between closely related strains but lower between more distant ones. Such a pattern can arise, for example, if homologous recombination is suppressed by mismatches between donor and recipient sequence [39]. Such models are thought to explain much higher

recombination rates within *Escherichia coli* phylogroups than between them [40]. Many models with these properties are possible, but since they all reduce the effectiveness of recombination, they are likely to make it difficult to recapitulate the low levels of LD in KP at large genetic distances. Moreover, the models that have been investigated previously cause species to cluster into discrete groups [41,42], rather than generating an approximately continuous pattern of variation. We are unaware of any model in which variable recombination rates within a single population predict a continuous structure similar to that seen in KP.

Genetic differentiation can arise due to geographical structure, or other factors that restrict gene flow between particular subsets of the population [23,29,43]. To explore the properties of geographical population structure we next simulated a stepping-stone model with 10 demes (Fig 6D). The demes are organized along a line, with a small number of migrants moving between adjacent demes each generation. Data simulated according to this model gave a tree with a phylogenetic backbone similar to that seen in KP, with PC1 explaining a similar fraction of the overall variation (Figs 6E, 6F, and S15) and also approximately matched the LD decay curve (Figs 6A and S16). However, the distribution of the SNP loadings for PC1 for these simulations did not match the real data, because there were many more SNPs with intermediate loading values (Fig 6G). Geographic population structure creates differentiation between populations through genetic drift, which is a random process that affects all of the genome, but to differing degrees. The discrepancy in SNP loading values between simulated and real data implies that the variation in differentiation values in KP is higher than expected under a pure random drift model.

A longer-tailed distribution of drift levels, and hence PC1 values, could be generated by natural selection at a fraction of loci associated with adaptation to specific demes. However, even if a geographic model with some selection could be made to fit the data, this model seems to be poorly motivated in the case of KP. There is no evidence of KP being structured geographically (Fig 1B) [10,21] or of there being physical barriers to gene flow between ecological demes, since, for example, it is readily acquired by humans from a wide variety of ecological sources [44–46]. It is therefore hard to see how a model based on strong isolation between demes could be applicable to the species.

A related possibility is that population structure might emerge from heterogeneity in mutation or recombination rates among strains, which has been proposed as another mechanism capable of generating deep phylogenetic divergence [47,48]. To evaluate this, we examined whether within-population heterogeneity in mutation or recombination rate could reproduce the backbone observed in KP. We simulated two alternative forms of heterogeneity. Introducing strong HGT hotspots, which locally increase the rate of genetic exchange, failed to produce any lineage structure and instead homogenized the population (S17A–S17C Fig). We next allowed strains within a single freely recombining population to vary in their mutation rate without introducing demographic structure. This model also failed to reproduce the backbone, the PC clustering, or the LD decay seen in the real data (S17D–S17F Fig). These results indicate that heterogeneity in recombination or mutation rates, whether localized in genomic regions or distributed among strains within the same population, is insufficient to account for the empirical patterns.

## KP variation is consistent with diversifying selection on a quantitative trait

Many traits are subject to diversifying selection [49]. For example, beak size has been shown to have two distinct optima in a species of Darwin's finch, *Geospiza fortis,* which seems to have arisen because birds can specialize either on small soft seeds or larger nuts. Birds with intermediate beak sizes are not well suited to either type of food source and have lower fitness during droughts [50].

To investigate the signatures of diversifying selection, we performed simulations of selection on a single quantitative trait in a recombining bacterial population, in which strains with extreme trait values have higher fitness, while the fitness function is flat for intermediate trait values (see Methods). Simulation results in which half of all new mutations increased fitness by one and a half decreased fitness by one showed that low recombination rates allow progressive divergence in trait values (Fig 7A–7C). At intermediate recombination values, a phylogenetic backbone appeared that is similar to the

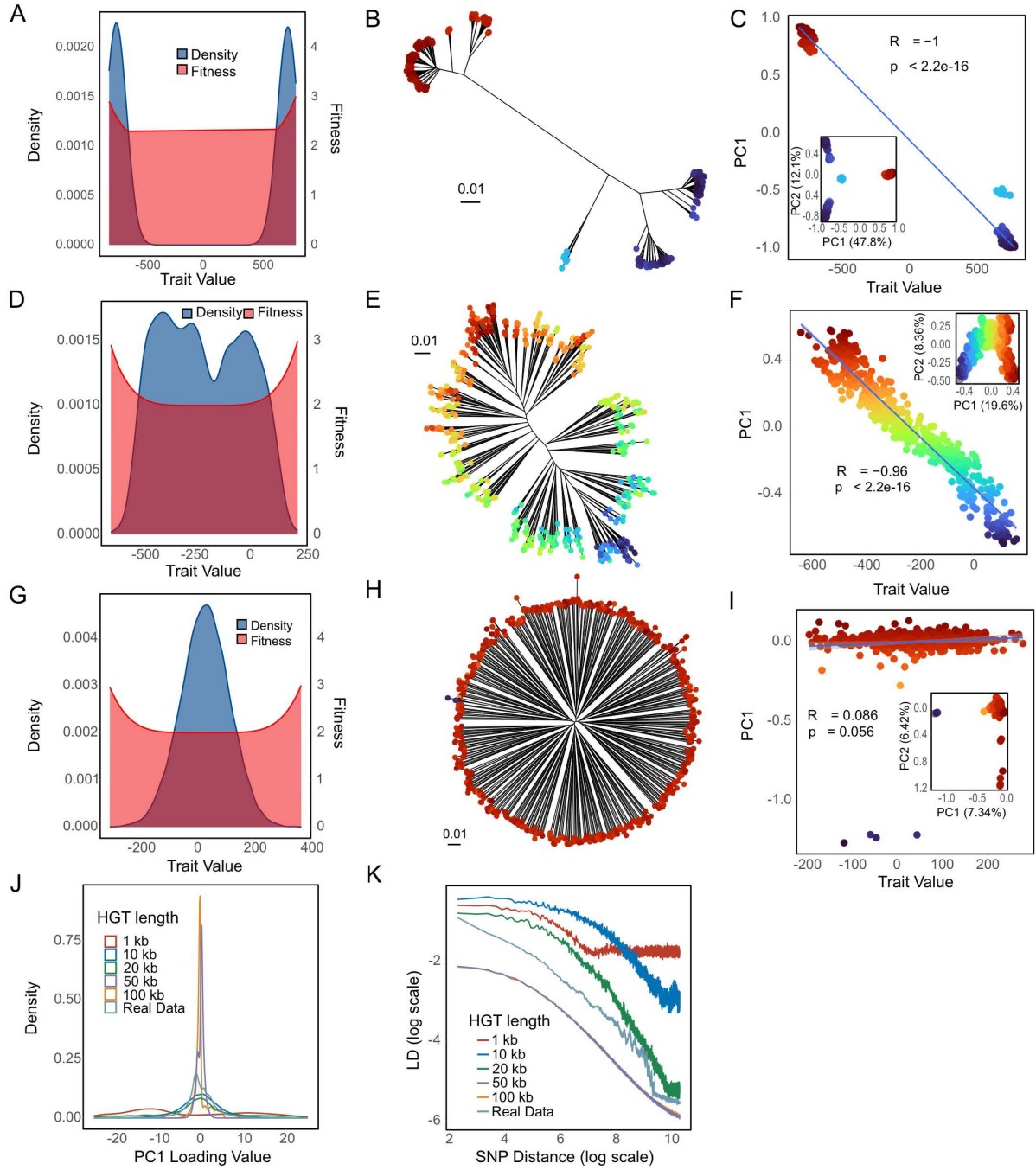

**Fig 7. Simulation of diversifying selection on a polygenic trait at different recombination rates. (A, D, G)** Trait value density distributions and fitness coefficients for recombination lengths of 1, 20, and 100 kb (top to bottom) in the final generation of simulation. **(B, E, H)** Phylogenetic trees depicting evolutionary relationships under each HGT length. **(C, F, I)** Correlations between PC1 values and trait values under each HGT length, with corresponding PC1 versus PC2 plots inserted in each panel. These simulations use 1Mb genomes. **(J)** Overall distribution of SNP loadings across all conditions. **(K)** Overall LD decay patterns across all conditions. The data underlying this Figure can be found in https://zenodo.org/records/18520201.

one seen in the KP data (Fig 7D–7F), while high rates homogenize the population, leading to a circular star-shaped phylogeny (Fig 7G–7I). For low and intermediate recombination rates (1, 20 kb), PC1 values were strongly correlated with the true simulated trait values (Fig 7C and 7F). As expected, LD values are strongly dependent on recombination rates (Fig 7K), with intermediate rates giving LD curves closest to the KP data.

These simulations failed to recapitulate the sharp peaks in SNP loadings seen in the KP data (Fig 7J). We therefore modified the simulations by assuming that the distribution of the magnitudes of the effect of SNPs on the trait followed a Gamma distribution, with shape and scale parameters controlling the skewness and overall magnitude of effect sizes, while holding the mean of the distribution fixed at 1 (Fig 8). High values of the Gamma scale parameter correspond to some mutations having a much larger effect on the trait than others (Fig 8A, 8D, and 8G). As the Gamma scale increases, the association between simulated PC1 scores and trait values remains strong, whereas the distribution of trait values becomes increasingly skewed toward more extreme values (Fig 8B, 8E, and 8H). In these simulations, each PC1 loading peak corresponds to or is immediately adjacent to a SNP with a large effect on trait values. For high Gamma values, the number of peaks is smaller (Fig 8C, 8F, and 8I).

We found that simulations with a very large Gamma value could recapitulate the sharp peaks seen in the KP data (Fig 8J). For this set of parameter values, there is a great deal of variation in PC1 loadings outside the sharp peaks, which is; however, not influenced by the SNP coefficients (Fig 8C, 8F, and 8I) and therefore represents noise, that is generated by genetic drift. For high Gamma shape parameters, the correlation between strain PC1 values and the trait values accordingly becomes weaker (compare Fig 8H with 8B and 8E), including if PC1 values are calculated only using top-loading SNPs. For intermediate Gamma shape parameters, the distribution of SNP loadings looked similar to that obtained assuming each SNP has the same effect on trait values (Fig 8J), but the distribution becomes more leptokurtic (i.e., with many SNPs with low values and long thin tails) as Gamma increases.

We found that a reasonable qualitative fit to the KP data could be found by simulating a 5 Mb genome length (similar to the true length of the KP genome) and assuming trait values for each SNP followed a Gamma distribution with a mean of 1 and a scale parameter of $5 \times 10^6$ (shape of Gamma distribution $= 2 \times 10^{-7}$) (Fig 9). Specifically, we were able to recapitulate a phylogenetic backbone (Fig 9A and 9B), a PC1 correlated with the trait value (Fig 9E) and a small number of high-loading peaks (Fig 9C), each of which is associated with SNPs with substantial influence on trait values. These simulations have slightly weaker LD than is seen in the real data (Fig 9H), with a smaller proportion of the variance explained by PC1. However, the pairwise LD between the highest loading SNPs showed a very similar pattern to that seen in real data (Fig 9D).

## Discussion

### Importance of quantitative traits in biology

Quantitative trait models are central to our understanding of human disease genetics [51] and animal [52] and plant [53] biology. These models are applicable to many morphological and disease susceptibility traits, which fit the assumptions of the model, namely that the trait value is dependent on many variants, none of which explain a substantial fraction of the overall variation. Furthermore, each variant acts approximately independently of the others in its influence on the trait value [54,55]. These features allow the traits of offspring to be predicted based on the values found in their parents. The predictability inherent in these models has found several compelling practical applications, including predicting disease risk in humans using Polygenic Risk Scores [56] and to assess the genetic quality of livestock, based on estimated breeding values [57]. The same rules also make short-term evolution relatively predictable [58]. Quantitative genetics can be used, for example, to predict the change in beak sizes in populations of Darwin's finches based on the intensity of selection and the additive genetic variation present within the population before selection takes place [50].

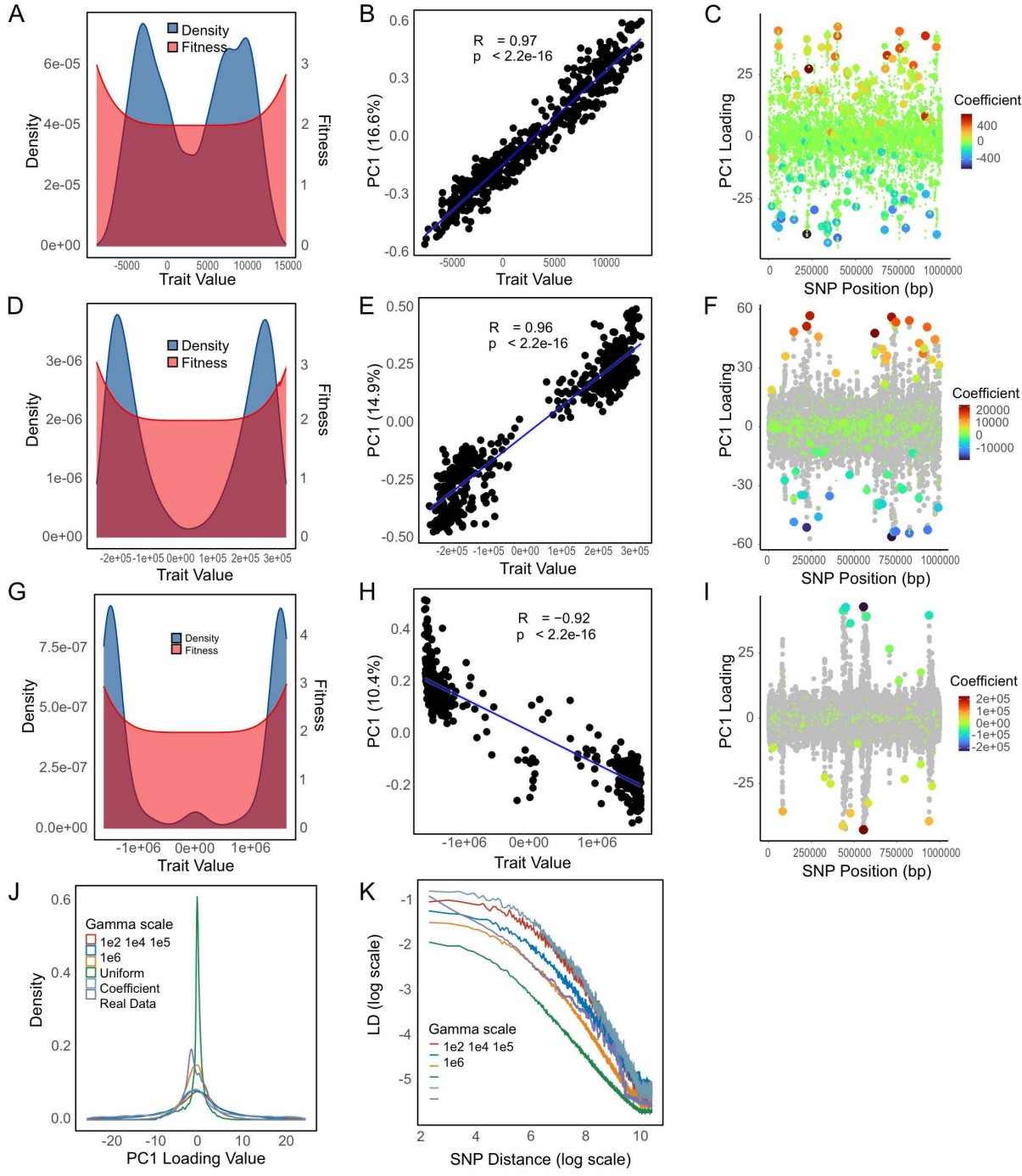

**Fig 8. Effects of different selection coefficient on diversifying selection. (A)** Distribution of fitness values under a gamma-distributed SNP effect size model (Gamma scale = 1e$^2$), plotted against trait values. **(B)** Correlation between trait values and PC1 scores. **(C)** SNP loadings along PC1, with point colors indicating the magnitude of SNP coefficients. **(D–F)** Same as panels a-c, but under a gamma distribution with Gamma scale = 1e$^4$. **(G–I)** Same as above, but with Gamma scale = 1e$^5$. **(J)** Overall distribution of SNP loadings across all conditions. **(K)** LD decay pattern. The data underlying this Figure can be found in https://zenodo.org/records/18520201.

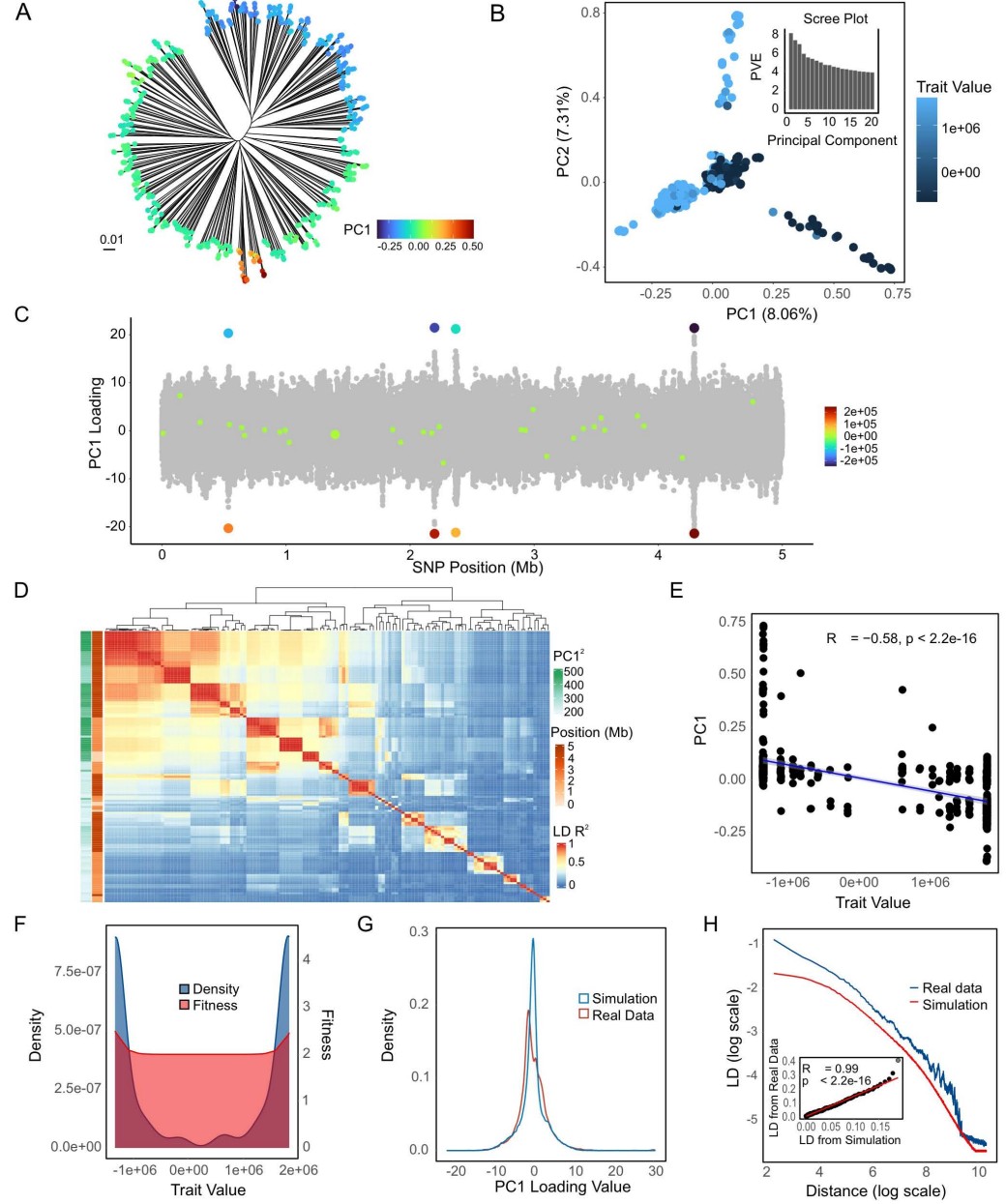

**Fig 9. Best-fit simulated diversifying selection model. (A)** Phylogenetic tree of simulated 5Mb genomes. **(B)** PCA results. **(C)** SNP loadings for PC1. **(D)** Heatmap of pairwise LD $R^2$ values for high-loading SNPs. **(E)** Trait value density distributions and fitness function. **(F)** Correlation between trait values and strain PC1 values. **(G)** Distribution of PC1 SNP loadings for the 5Mb simulation compared to the real data. **(H)** LD decay curves for the simulated and real datasets. An inset scatter plot compares the LD values between the two datasets, highlighting their correlation. The data underlying this Figure can be found in https://zenodo.org/records/18520201.

Bacteria do not undergo sexual reproduction in each generation, but instead reproduce clonally, while occasionally importing tracts of DNA from other strains [59]. The population-level response to selection depends on specific lineages rising in frequency [60]. As a result, the most straightforward applications of quantitative trait models in eukaryotes, such as predicting offspring trait values, do not apply [61]. Despite these differences, bacteria do have many continuous-valued

traits [62], such as cell size, growth rate, and antibiotic minimum inhibitory concentration (MIC). It is of interest to understand the genetic variation of each trait circulating within species and how the trait responds to selection.

Laboratory selection experiments have provided important evidence on the genetic architecture of the response to selection in a novel environment [63]. Bacterial fitness in a particular environment is a continuous-valued trait. In asexual laboratory populations, mutations of large effect dominate the initial response, but diminishing returns epistasis sets in rapidly with later variants having progressively smaller effect [64]. Thus, in these experiments, fitness does not fit the assumptions of quantitative trait models. However, although these laboratory experiments provide excellent evidence on responses to novel environments for clonal populations, we currently have neither data nor theoretical models applicable to natural recombining populations of bacteria. For example, if a lineage is selected for increased cell size, we do not know whether the response is likely to be driven by variants of small or large effect, and whether the variants will principally arise de novo or be imported from other members of the population by homologous or non-homologous recombination.

To our knowledge, the only continuous valued trait whose variation has been studied extensively in bacterial populations is antibiotic MIC, which can be treated quantitively [65]. Antibiotic resistance is, however, an unusual trait, because of the episodic nature of exposure to antibiotics and the strong selection coefficients involved, which accentuates the role of variants of large effect [66]. The most important sources of variation are mobile genetic elements carrying antibiotic resistance islands and de novo point mutations [67,68]. These variants also do not fit the assumptions of quantitative trait models because individual variants have a large effect and, moreover, there are strong epistatic interactions between them [69,70]. In any case, when antibiotic selection pressure is applied, instead of increasing MIC across the population, the outcome is some lineages become highly resistant, while others remain susceptible [71].

One factor that is likely to be critical in determining the genetic architecture of a trait is the past selection history [72]. For example, if cell size has previously been consistently selected to be smaller within the natural environment, then we might expect the response to selection for larger cell size to be like that seen in laboratory selection [73], with de novo mutation dominating and the initial response generated by mutations of large effect. By contrast, if there has been fluctuating or diversifying selection in the population, then many variants affecting size may already be segregating. These existing variants will have already been tested by selection and will tend to be less disruptive to other features of the organism's biology. Therefore, if we want to find traits that are polygenic within bacterial populations, where several variants behaving additively and most variants having small effect, we might focus on traits that have been under such selection regimes.

## Diversifying selection can generate bacterial ecoclines

Here, we have used simulation to demonstrate that there are circumstances in which selection on a quantitative trait might structure the genetic variation across an entire bacterial species. We performed simulations of an additive trait in which strains with the highest and lowest values of the trait have a small fitness advantage. In the simulations this tendency is counteracted by recombination, which randomizes variation within the population. If the recombination rates at each locus are low, strains are likely to differentiate into two discrete groups with distinct trait values (Fig 7A–7C). If recombination rates are higher, then intermediate strains can become common (Fig 7D–7F) and variation becomes clinal. Because this cline is generated by selection on a trait, we call it a bacterial ecocline.

The clinal variation we see in our simulations depends on a balance between selection and recombination and can potentially be seen throughout the geographical range of the species. As such it is distinct from the geographic clinal variation commonly seen in outbreeding animals and plants. Eukaryotic biologists use the term ecocline to describe a situation where there is a single key ecological variable, such as altitude above sea levels for terrestrial animals or saltwater concentration for marine ones, which generates a gene frequency cline in the loci that determine adaptation to that variable [74,75].

Sexual reproduction randomizes variation faster than bacterial genetic exchange, so eukaryotic clines are normally dependent on limited dispersal between locations, allowing differentiation in trait values, which in turn reduces genetic exchange between individuals with high and low trait values. We think that this form of variation should strictly be called an eco-geocline, because it is necessarily mediated by geography. However, by calling it an ecocline researchers implicitly hypothesize a central role for an ecological trait, and in this sense, while the genetics and population genetics details differ between prokaryotes and eukaryotes, the underlying idea behind naming it an ecocline is similar, namely that selectively driven differentiation drives clinal variation.

Fitness tradeoffs that might drive differentiation, such as between fast growth and starvation resistance, are important for all bacterial species. However, given our limited understanding of selection in natural bacterial populations, it can be difficult to predict which traits might be subject to diversifying selection. To identify them, we therefore suggest a "reverse ecology" approach [23,28,76,77] in which we first identify clinal genetic variation in bacterial species, before seeking to identify the associated trait or traits and then seeking to understand why the trait is under selection. This approach led us to a detailed analysis of genetic variation in KP.

### Clinal variation in KP

The KP phylogenetic tree is structured by a backbone, which has been similarly informally noticed previously by several researchers but resists immediate explanation. To characterize the underlying variation in more detail, we used PCA, which is better suited to characterizing continuous variation than phylogenetic methods. When PCA is applied to a non-redundant dataset, PC1 is strongly correlated with the backbone. However, caution must be exercised in interpreting PCA results, because the method will detect apparently clinal variation in essentially any dataset. The PC1 cline in KP is noteworthy because of the large fraction of variation it explains (Fig 1B), its reproducibility in analyses using subsets of the data (Fig 2) and the absence of geographic differentiation in the species.

Our PCA results are particularly useful because they give us the first clues about the possible genetic architecture of the clinal variation. Specifically, PC1 variant loadings have four prominent peaks, corresponding to Linkage Disequilibrium (LD) blocks in the genome. In the simulations of selection on a quantitative trait that best fit these data, the Gamma parameter determining variation in trait values is large, meaning that most of the variation is generated by a small number of loci. In these simulations, we found that the highest PC1 loading peaks corresponded to the loci with the largest effect on trait values, but that remaining variation in PC1 appears to be approximately independent of trait values, i.e., there is little discernable difference between sites that had a moderate or small effect on trait values and completely neutral sites (Fig 8).

We have also visualized the LD between high-loading regions of the genome in KP (Fig 4D) and found a pattern that is broadly consistent with the patterns seen in a comparable simulated ecocline (Fig 9D). In the simulations, selection generates substantial LD blocks centered on each of the loci that have large effects on trait values. These blocks are in LD with each other, generated by diversifying selection. In the KP data, block 1 has the most extreme PC1 loadings, so interpolating from our simulations, this block is likely to have the largest effect on the putative quantitative trait. Concordant with this prediction, it has consistently high pairwise LD with loci in other blocks, while blocks 5–44, which have lower loading values, have consistently low pairwise LD values with other blocks. The LD patterns that we see thus seem broadly consistent with selection being mediated by the effect on a trait, rather than direct pairwise interactions between loci, which would generate a more patchwork-quilt-like pattern.

### Testing the KP ecocline hypothesis

Our comparison with simulated data implies that diversifying selection on a trait is a plausible explanation for the phylogenetic backbone, and the concordant signal in PC1, seen in KP. However, to demonstrate that this explanation is correct, we will need to come to an understanding of how selection is operating. A first step will be to demonstrate functional

coherence of the loci with extreme trait values. The ecocline hypothesis specifically implies that each of these loci should be altering the same ecologically relevant trait in a consistent way. Once this consistency has been demonstrated, further steps will be to show that the trait is under diversifying selection, and that this selection is sufficiently strong across the range of the species to maintain the variation in trait values, counteracting the homogenizing effect of homologous recombination.

Existing functional annotations of the genes in the top four blocks provide hints, but not strong evidence that the genes might influence the same trait. The most differentiated gene, Block 1 in Fig 4A, is the adhesin component of Kpa fimbriae, which suggests the trait may be related to adhesion/diffusion. Block 2 encodes a filamentous hemagglutinin, which may also be an adhesin. The EAL- and GGDEF-domain proteins encoded in blocks 3a and 3c, respectively, are related to c-di-GMP metabolism, a second messenger that controls trade-offs between sessile/surface and motile/free-living behaviors [78]. The accessory *kfuABC* genes in block 3a suggest there is some involvement of iron acquisition [36]. While these genes were previously found to be over-represented in invasive clinical isolates and bovine mastitis strains [35], we could not find any evidence that the trait is directly and specifically involved in human infection. One thing that is clear is that virulent or resistant genotypes can assemble on all positions of the ecocline.

We also attempted to identify specific gene function categories that were particularly associated with the backbone. For core genes, there was no difference between COG categories in mean absolute PC1 loading values (Fig 5). These results are consistent with our simulation results, which imply that if a trait is determined principally by a small number of loci, then genes that modify the trait by smaller amounts cannot be distinguished from neutral loci based on PC1 values. Therefore, it seems likely that outside the four peaks, most of the variation in PC1 values in core genome SNPs is generated by genetic drift. However, for accessory gene presence-absence, metabolic and cellular process-associated genes have systematically higher absolute loadings than genes of unknown function or those associated with replication, recombination, or transcription, which provides suggestive evidence that these are under stronger selection.

Despite the provisional nature of the current evidence, we have been emboldened to propose KP variation might be organized into an ecocline based on our previous finding of "ecospecies" in two other bacterial genera [23,28,29]. An ecospecies is like an ecocline, in that genetic differentiation is maintained by selection in the absence of other barriers to genetic exchange. However, in contrast to bacterial ecoclines, each strain belongs to one ecospecies or another, and there are no intermediates. For example, differentiation between Hardy and Ubiquitous *Helicobacter pylori* is largely restricted to 100 genes, which each have nearly fixed differences in at least 5 SNPs and there are no intermediate genotypes [29]. In VP, differentiation between Molassodon and Typical strains has taken place at 40 core genes, and once again there are no intermediate genotypes [23,28].

For ecospecies, we have also been able to make stronger functional predictions, relating the VP ecospecies to motility [23] and the *H. pylori* ecospecies to host diet [29]. We have also recently used laboratory experiments to confirm and extend the original functional predictions and show that the differentiation between VP Typical and Molassodon ecospecies is functionally coherent, with the Molassodon likely having evolved a hunt, kill, and devour phenotype that it employs in viscous liquids [28]. These results show that the reverse ecology approach can identify loci that act together in natural populations.

The absence of equivalent functional predictions means that our current designation of a bacterial ecocline in KP is provisional and needs to be validated by phenotypic data. We have used simulations to explore other hypotheses, for example, including clonal structure and geographical barriers and not found any single explanation for the data that gives a comparable fit to the data as simple bacterial ecocline. However, it is impossible to rule out more complex scenarios.

If the ecological hypothesis is correct, and the genes with extreme loadings are influencing a particular ecological strategy, then characterization of the effects of these variants should simultaneously provide considerable insight into gene function, ecology, and evolution for a key bacterial fitness trait. This would likely represent a breakthrough as these aspects are hard to study individually, handsomely justifying the initial effort required to identify the trait.

It should be cautioned, however, that diversifying selection may not reflect a single laboratory-measurable trait. For example, distinct foraging strategies in Darwin's finches might require different beak sizes but also different behaviors, which may also be genetically determined. This could create LD between genes determining beak size and foraging behavior. This ecologically-generated LD is also evident in the Molassodon ecospecies example [23,28], where genes responsible for motility, bacterial killing, and nutrient uptake have coevolved. It thus might be hard to find a simple laboratory condition that links together the function of the different genes involved in either ecospecies or ecoclines.

A challenge and virtue of the reverse ecology approach [28,76,77] is therefore that it confronts us with the gulf between laboratory conditions and the natural environment that we seek to understand. In any case, we propose that investigation of phenotypic differences associated with the PC1 cline has the potential to deliver key insights into KP ecology, as well as on the biology of quantitative traits in microbes.

## Methods

### Genome collection and sequence alignment

We collected 1,421 KP genomes from the NCBI database, following the reference dataset provided by iterative-PopPUNK [79], along with a reference genome (NTUH-K2044, NC_012731.1). Additionally, we obtained genome metadata using the NCBI Datasets command-line tools (v16.12.1) [80] and used Kleborate (v2.4.0) [24] to extract clinically relevant genotypic information on strains. All genomes were aligned to the NTUH-K2044 reference genome using Nucmer (v3.1) [81] for reference-based alignment.

In total, 8,331 VP genomes were downloaded from the NCBI database, following the dataset used in the study 'Why panmictic bacteria are rare' [82]. All genomes were aligned to the reference strain VP RIMD 2210633 (RefSeq NC_004603.1 and NC_004605.1) [83] using Nucmer (v3.1) [81], following the approach described in previous VP studies [22,23].

### Core genomic variant calling and definition of non-redundant dataset

SNP-sites (v2.5.2) [84] was used to call all the variants from the whole KP dataset alignment. Core SNPs were defined as the SNPs present in at least 99% of the genomes across the whole genome alignment, resulting in 850,158 core SNPs after extraction. Using these core SNPs, pairwise distances were calculated across all genomes with snp-dists (v0.8.2) [85]. A non-redundant dataset of 369 genomes was defined by selecting genomes with pairwise SNP distances >20 kb. The corresponding metadata, including CGF ID (Genome Context File ID), host type, isolation source, and geographical origin, are summarized in S2 Table.

The same method to the whole VP dataset, using a 50 kb SNP distance threshold to reduce redundancy, yielding 667 genomes (S3 Table) with 1,619,597 core SNPs.

### fineSTRUCTURE and population assignment

To further investigate the population structure of the non-redundant 369 KP strains described above, we performed chromosome painting and fineSTRUCTURE analysis. Using ChromoPainter mode in fineSTRUCTURE (v4.1.0) [18], we inferred chunks of DNA donated from a donor to a recipient for each recipient haplotype and summarized the results into a co-ancestry matrix. Subsequently, fineSTRUCTURE clustered 369 individuals based on the co-ancestry matrix, with 100,000 iterations for both burn-in and Markov Chain Monte Carlo. In this analysis, only variants present in all non-redundant strains were considered, as fineSTRUCTURE requires input data without missing values.

To assign a population to the non-redundant VP dataset, we followed the same fineSTRUCTURE approach using haplotype data from core SNPs. Based on the fineSTRUCTURE results, the genomes were classified into five populations: VppAsia (506 strains), Hybrid (7 strains), VppUS1 (50 strains), VppUS2 (30 strains), and VppX (74 strains).

## Construction of phylogenetic trees

Based on the core SNPs defined above, we constructed nucleotide sequence alignment files for both the entire KP dataset and the non-redundant KP dataset. Phylogenetic trees were constructed using FastTree (v2.1.10) [20] with the "-nt" option under the default model. Phylogenetic reconstruction was additionally performed with IQ-TREE (v2.0.3) using maximum likelihood. We used ModelFinder to select the best-fit substitution model (option -m TEST) and assessed branch support with 1,000 ultrafast bootstrap replicates (-bb 1,000) and 1,000 SH-aLRT replicates (-alrt 1,000). The trees were visualized and compared using the ggtree package (v3.12.0) [86] in R.

Similarly, we used the same software and parameters to construct phylogenetic trees for the non-redundant VP and VppAsia datasets, as well as the simulated datasets. During Partial-Variants Matching, we also constructed phylogenetic trees based on a subset of SNPs from the real datasets.

## Principal component analysis

Before performing PCA (Principal Component Analysis) on the non-redundant KP dataset, PLINK (v1.9) [87] was used to filter out loci with a minimum allele frequency (MAF) below 2% from the core SNPs, leaving 102,226 variants. LD pruning was subsequently performed to remove linked SNPs, where one SNP from each pair was removed if the LD ($R^2$) exceeded 0.1 within a 50 kb window. The window was then shifted forward by 10 bp, and the procedure repeated to ensure comprehensive coverage. A total of 33,589 pruned variants were obtained and used for conducting PCA on the non-redundant dataset with PLINK [87], extracting and reporting the first 20 principal components (PCs) by default.

Similarly, we used the same software and parameters to perform PCA on the non-redundant VP dataset, which included 220,737 SNPs after MAF filtering and 69,537 SNPs after pruning. We also applied PCA to the VppAsia dataset, containing 207,292 SNPs after MAF filtering and 65,058 SNPs after pruning. For the simulated datasets, we only applied MAF filtering to retain enough SNPs, followed directly by PCA without pruning.

## Variant loadings calculation

Variant loadings represent the contribution of each genomic locus (core SNPs or accessory genes) to a given principal component. In particular, PC1 loadings describe how strongly variation at each locus contributes to the main axis of genetic differentiation among strains. In this study, we first calculated the correlation between the original variables and the PCs, referred to as the PCA loading for strains. The loadings can be determined as the coordinates of the variables (eigenvectors) multiplied by the square root of the eigenvalues associated with that principal component (Equation 1). When performing PCA, PLINK [87] generates the corresponding eigenvalue (.eigenval) and eigenvector (.eigenvec) output files. Next, we transform the MAF-filtered SNP matrix into a binary matrix, assigning a value of 1 to minor alleles and 0 to all others (including the third and fourth most frequent alleles at non-biallelic sites). Finally, through matrix multiplication, we computed the product of the core SNP matrix and the PCA loading matrix to obtain the loading values for the MAF-filtered SNPs (Equation 2).

$$strainloadings = eigenvectors \cdot \sqrt{eigenvalues}$$

(1)

$$SNPloadings = strainloadings \cdot SNPmatrix$$

(2)

The absolute PC1 loading values were calculated as the absolute values of the PC1 loadings, representing the strength of association between each genomic locus and the principal axis of genetic differentiation. The squared PC1 SNP loading values were then used to estimate the proportion of variance in PC1 explained by each locus, providing a direction-independent measure of their contribution to the main axis of genetic variation.

To better highlight the significant peaks in the non-redundant KP result, we plotted the squared PC1 SNP loading values and marked the dots with squared loading values greater than 300.

Similarly, we used the same method to calculate SNP loadings for the non-redundant VP dataset, the VppAsia dataset, and the simulated datasets.

## Pangenome analysis

To estimate the gene content, we annotated all KP genomes with prokka (v1.13) [88] software, using a curated version of the NC_012731.1 (NTUH_K2044) proteome as the primary annotation source. The Generic Feature Format (GFF) outputs generated by prokka were then used as the input files for panaroo (v1.2.8) [30] with the strict model. The "--remove-invalid-genes" and "--merge_paralogs" options were also employed during the analysis. Next, we extracted the non-redundant dataset from the overall results and defined 5,558 accessory genes as those present in between 2% and 98% of the genomes in the non-redundant dataset. Using the same approach (Equation 2), we also calculated the gene loadings and defined genes with squared PC1 loadings greater than 300 as high-loading genes.

## Calculation of linkage disequilibrium decay

LD refers to the non-random association of alleles at different loci within a population. Assessing LD patterns provides insight into the extent of recombination and the historical relationships among genomic sites. To analyze the decay of LD in KP, we included 102,226 MAF-filtered SNPs and used the Haploview (v4.2) [89] software to compute $R^2$ values, where $R^2$ represents the squared correlation coefficient between alleles at two loci and is a standard measure of LD. When running Haploview, we set "-maxdistance 30" to ensure that the maximum marker distance for $R^2$ calculation did not exceed 30 kb. Additionally, we set "-minMAF 0" and "-hwcutoff 0" to exclude any filtering based on minor allele frequency and Hardy–Weinberg equilibrium testing, ensuring that all input SNPs were included in the calculations. The average $R^2$ value was calculated using a 10 bp step size.

Similarly, the same software and parameters were used for the simulated datasets to obtain the LD decay results for the simulated data.

We also calculated the LD between the high-loading variants defined above (340 variants in total, including core SNPs and accessory genes) used Equation 3 in R. Each element $R_{ij}^2$ of the matrix quantifies the degree of LD between variant $i$ and variant $j$, providing a comprehensive view of the genomic associations among core SNPs and accessory genes. Based on manual inspection, we removed an accessory gene (group_205) that represented fragmented versions of KP1_1982 (339 variants left), which is visualized in a heatmap, generated using the ComplexHeatmap package [90].

$$R^2 = \frac{\left(f_{gene_i,gene_j} - f_{gene_i}f_{gene_j}\right)^2}{f_{gene_i}f_{gene_i'}f_{gene_j}f_{gene_j'}}$$

(3)

Based on the LD heatmap of all high-loading variants, we performed hierarchical clustering to divide the heatmap into 44 blocks. Among these, four larger blocks contain multiple variants, while the remaining 40 smaller blocks each contain only a few variants. Notably, block 3 was further subdivided into 3a, 3b, and 3c based on the physical locations of the mutations on the reference genome. We observed that some SNPs in block 3 are located within the KP1_RS30360 and KP1_RS30365 genes in the reference genome. According to the annotation file of the reference genome, KP1_RS30365 is classified as a pseudogene. However, we do not concur with this interpretation. We believe that KP1_RS30360 and KP1_RS30365 together represent a complete gene, with some strains (*e.g.*, MGH 78578) containing only a fragment of this gene. The differentiation at this locus also likely represents a functional presence/absence rather than functional differentiation via SNPs.

As mentioned above, we converted the variants into binary, where 0 indicates that the core SNP is a major SNP and the accessory gene is absent in the strain, and 1 indicates that the core SNP is a minor SNP, and the accessory gene is present in the strain. We then generated heatmaps of the variant states according to the blocks, with rows following the same order as the LD heatmap and columns subjected to hierarchical clustering.

## Phylogenetic reconstruction of a gene across multiple *Klebsiella* subspecies

Genomes were retrieved from the Pasteur Institute *Klebsiella* MLST Database by applying the following filtering criteria: (i) the taxonomic designation must be one of the following subspecies: *K. quasipneumoniae subsp. quasipneumoniae*, *K. quasipneumoniae subsp. similipneumoniae*, *K. variicola subsp. variicola*, *K. variicola subsp. tropica*, *K. quasivariicola*, or *K. africana*; (ii) the annotation status for MLST, ribosomal MLST, and scgMLST629_S must all be labeled as good; and (iii) assembly quality checks including the number of contigs, assembly size, minimum N50, and %GC must all pass.

To confirm species identity, the resulting genomes were further screened using Kleborate [24]. The gene KP1_0338 from the reference strain was used as a query in a BLAST search, and MGH 78578 was included as an additional representative strain. To avoid overrepresentation of any subspecies, 10 genomes were randomly selected from each subspecies identified in the BLAST results.

These selected genomes, together with the reference strain, MGH 78,578, and the 5 strains with the highest and 5 strains with lowest PC1 loadings from the previously defined non-redundant dataset, were aligned using MAFFT [91]. The resulting multiple sequence alignment was then used to construct a phylogenetic tree using FastTree [20].

## Functional annotation

To further characterize the functional content in KP, we annotated both the MAF-filtered SNPs (MAF > 0.02) with SnpEff (v4.3t) [92], using the reference genome NTUH-K2044 (NC_012731.1), and applied custom filtering to focus on coding region variants by disabling annotation for upstream, downstream, and UTR (untranslated) regions. We then obtained the types of the SNPs and determined whether they were synonymous mutations. For accessory genes, we used the eggNOG-mapper [93] online tool with its default settings to ensure a comprehensive and standardized annotation.

For the identified high-loading variants (squared PC1 SNP loading values > 300), we further mapped these variants to another reference genomes(MGH 78578, NC_009648.1) using BLASTn [94]. The mapping aimed to identify their genomic locations and potential functional implications in these strains. Additionally, we searched the UniProt [95] database to retrieve gene names and associated functional annotations for the high-loading variants.

We calculated the mean absolute PC1 loading for accessory genes within each COG category, as well as the overall mean absolute loading across all accessory genes. For core SNPs, we first computed the mean absolute PC1 loading of SNPs belonging to each core gene, then calculated the average of these values for each COG category. In addition, we determined the overall mean absolute loading across all core genes.

We visualized the results using bar plots, where each bar represents the mean absolute PC1 loading of genes (core or accessory) within a given COG category, and the error bars indicate the standard error of the mean. A solid line denotes the overall mean absolute PC1 loading across all accessory genes (or core genes), serving as a reference.

## Isolate background characteristics

To investigate potential host specificity in KP, and to address the overrepresentation of human-derived isolates in our primary dataset, we incorporated a multi-host dataset from the Pasteur Institute *Klebsiella* MLST Database (https://bigsdb. pasteur.fr/cgi-bin/bigsdb/bigsdb.pl?db=pubmlst_klebsiella_isolates). Isolates were selected based on the following quality criteria: chromosomal assembly size between 5 and 6 Mb, fewer than 100 contigs, an N50 greater than 20,000 bp, fewer than 1,000 ambiguous bases (Ns), and a GC content between 55% and 60%. Additionally, only genomes with "good"

annotation status for MLST, ribosomal MLST, and SCGMLST629_s schemes were included. Kleborate [24] was applied to refine species classification, and *Klebsiella pneumoniae* species complex (KpSC) isolates were excluded. To minimize overrepresentation of human-associated strains, genomes with host information explicitly annotated as "human" were excluded. However, a small number of human-derived isolates may still be present due to inconsistent or alternative host labels (e.g., *Homo sapiens*). (S4 Table).

We used the KP1_0338 gene from the reference strain as the query sequence and included MGH 78578 as an additional representative. These two strains represent the extremes along the PC1 axis in our population structure analysis, with NTUH-K2044 showing an extremely high PC1 value and MGH 78578 a low one, thus capturing the two major variants of KP1_0338 present in the KP population. Using BLASTn [94], we identified KP1_0338 homologs in the multi-host dataset. The retrieved sequences were aligned using MAFFT [91], and a phylogenetic tree was constructed with FastTree [20] to examine sequence diversity. Based on the topology of the KP1_0338 phylogeny, isolates were grouped into two distinct clusters, designated as High and Low types.

To assess host-specific enrichment of these gene variants, we performed a one-vs-rest Fisher's exact test for each host. This test compared the counts of High and Low types within each host to those in all other hosts combined.

We projected the combined dataset, including the non-redundant dataset as well as strains isolated from water and horse sources, onto the principal component space defined by the non-redundant dataset. For each group (non-redundant, water-derived, and horse-derived), we calculated the mean PC1 loading values and visualized them with standard deviations as error bars. Pairwise comparisons between groups were performed using two-sample t-tests to assess the statistical significance of differences in PC1 loadings.

## Half-matching testing

To assess the stability and repeatability of PC1 in KP, we used three half-matching tests, each designed to evaluate the consistency of PC1 under varying conditions.

**1. Half-strain matching.** Firstly, we divided the non-redundant KP dataset into two groups based on the backbone in the phylogenetic tree. We then performed PCA on the first group using the pruned SNPs. Afterward, we calculated the MAF-filtered SNP loadings using the above steps. Subsequently, we obtained the product for the second group through matrix multiplication, representing the mapping of the PCA results from the first group onto the second group. We then calculated the correlation for the PC1 between the mapped results of the first group and the PCA results of the second group. Conversely, the same matching analysis was performed on the data from the second group.

**2. Half-genome matching.** KP has a circular chromosome with a length of approximately 5–6 Mb, and the reference genome we used is approximately 5.2 Mb in length. We divided the circular chromosome into two DNA fragments (Fragment 1 (1 bp–2.6 Mb) and Fragment 2 (2.6 Mb–5.2 Mb)), each containing approximately half of the chromosome. Specifically, the first dataset comprised the pruned SNPs in the first half of the chromosome, while the second dataset encompassed the unlinked SNPs from the second half. We separately performed PCA on these two datasets using PLINK [87]. Finally, we calculated the correlation between the PC1 of the two fragments.

**3. Partial-variants matching.** First, we assigned maximum loading values to the MAF-filtered SNPs based on their squared PC1 loadings, using a 50 kb window size with a 1 bp step size. This process identified the maximum squared PC1 loadings within each 50 kb vicinity. Based on the distribution of maximum squared PC1 loadings, the SNPs were categorized into three types: low-loading SNPs, medium-loading SNPs, and high-loading SNPs. We separately conducted PCA on the high-loading SNP dataset and low-loading SNP dataset, by extracting unlinked SNPs from each. Subsequently, we calculated the correlation between the PC1 of the 2 datasets and constructed the phylogenetic tree for the 2 datasets. In this test, we ignored SNPs with intermediate loading values.

Similarly, the three types of half-matching tests were applied to the non-redundant clonal population simulation (Clonal Haploid Bacteria Model with Horizontal Gene Transfer (HGT), SNP distance > 2 kb, shown in S4A Fig), the non-redundant

VP dataset and the VppAsia dataset. However, in the half-strain matching test, although we still performed splits along the phylogenetic tree, these datasets were not always divided into left-hand and right-hand subsets in the trees. Therefore, in these datasets, we referred to the two subsets as half samples 1 and half samples 2. Additionally, in VP, which contains two chromosomes, the half-genome matching is implemented by separating the data according to chromosome number rather than splitting each chromosome into two segments.

## Simulation of bacterial populations

We used SLiM (v4.0.1) [96] to simulate the evolutionary dynamics of bacterial populations under different scenarios using a non-Wright–Fisher (nonWF) framework [97]. All scenarios started with identical initial conditions: each bacterial genome consisted of a single 1 Mb haploid chromosome, with a mutation rate of $10^{-6}$ per site per generation, approximately 10 times higher than the estimated mutation rate for [98]. HGT occurred at a rate of 1.0, ensuring that every individual experienced HGT in each generation. The simulations ran for 10,000 generations. Two mutation types, "m1" and "m2", occurred at each site with equal probability (1:1) and were assumed to be neutral. However, in the final simulation, we adjusted the chromosome length to closely match the length of a real KP genome.

## Modeling clonal haploid bacteria with horizontal gene transfer

In the first simulation scenario, we modeled the evolutionary dynamics of a single bacterial population, focusing on the spread of a single type of neutral mutation and the genetic exchange within population via HGT. The HGT tract length was set to 100 bp, 1 kb, 10 kb, or 100 kb, representing a range of recombination sizes to accurately capture genetic variation within the population. Each bacterium produced offspring according to a Poisson distribution with $\lambda = 2$, meaning that, on average, each bacterium produced two offspring per generation. However, the number of offspring per bacterium in any given generation could vary, with some bacteria producing more and others producing fewer, reflecting the probabilistic nature of the Poisson distribution. We conducted three trials with population sizes of 1,000, 2,000, and 10,000.

## Linear stepping-stone model

In the second simulation scenario, we constructed a linear stepping-stone model with 10 subpopulations of clonal haploid bacteria, where each subpopulation was connected by migration only to its nearest neighbors in the chain. The average recombination tract length within each subpopulation was set to 10 kb. Migration occurred with varying numbers of migrants—1, 10, 20, or 50 individuals—moving to neighboring subpopulations each generation to simulate gene flow driven by geographical factor or other isolating barriers. Each individual produced offspring according to a Poisson distribution with $\lambda = 2$. We conducted three trials with total population sizes of 1,000, 2,000, and 10,000, corresponding to 100, 200, and 1,000 individuals per subpopulation, respectively.

## Heterogeneity models

In the recombination-rate heterogeneity model, all individuals belonged to a single population and produced offspring according to a Poisson distribution with $\lambda = 2$. Homologous recombination occurred through 20 kb HGT tracts. To impose strong variation in recombination rate across the genome, 80% of HGT events were forced to initiate at one of three predefined hotspot positions (250, 500, and 750 kb), whereas the remaining 20% began at uniformly random positions. This scheme generated localized regions with markedly elevated recombination frequency while keeping all other evolutionary processes uniform across the genome.

In the mutation-rate heterogeneity model, we used two subpopulations (p1 and p2) purely as labels within a single freely recombining population. Both sets of individuals experienced identical demography and HGT-based recombination, but de novo mutations were disabled in p1, such that only p2 accumulated new mutations. Each individual produced

offspring according to a Poisson distribution with $\lambda = 2$, and HGT was modeled by copying a single 20 kb tract from a randomly chosen donor (50% from p1 and 50% from p2) into the recipient genome. Despite the internal labels, the population therefore functioned as a single panmictic group with persistent individual-level variation in mutation rate.

## Selective model

In the fourth simulation scenario, we focused on a population of 10,000 clonal haploid bacteria, where generations 1–5,000 evolved neutrally, and selective pressure was introduced from generations 5,001–10,000. HGT tract lengths were set to 1, 10, 20, 50, and 100 kb. In this model, the coefficient for any locus on the chromosome was 1. Depending on the mutation type, if it was $m1$, the value at that locus was 1; if it was $m2$, the value at that locus became −1. Each strain was assigned a score, termed a quantitative trait value, based on the difference in the number of "$m1$" and "$m2$" (score = $\sum m1 - \sum m2$), and fitness was determined by a normalized power function of the deviation from the mean score ($fitness = 2.0 + 1.0 \times (|score - \mu|^5) / (max(|score - \mu|^5) + 10^{-6})$, where $\mu$ is the mean score across all individuals). Based on the trait value–derived fitness (ranging from 2.0 to 3.0), individuals with more extreme trait values had higher fitness. The number of offspring produced by each individual followed a Poisson distribution with $\lambda = fitness$, such that individuals with extreme trait values tended to produce more offspring on average. This setting imposed a diversifying selection pressure, favoring individuals at the tails of the trait value distribution.

## Refined selective model

We refined the selective model by introducing locus-specific selection coefficients drawn from a Gamma distribution, which yields only positive values and allows specification of the mean. The mean was set to 1 to match the previous model, where all loci had equal effects of 1, thereby ensuring comparable overall selection intensity while allowing locus-specific variation. Since each locus had a specific selection coefficient, the assignment of values for each simulated strain was modified accordingly. The quantitative trait value was calculated as follows:

$$\text{score} = \sum_{i=1}^{N_{m1}} (m1_i \cdot \text{coefficient}_i) - \sum_{j=1}^{N_{m2}} (m2_j \cdot \text{coefficient}_j)$$

(4)

In this formula, $N_{m1}$ and $N_{m2}$ represent the number of $m1$ and $m2$ mutations in the simulation, respectively. This calculation differs from the previous simulations, where the trait value was simply the difference between the numbers of "$m1$" and "$m2$" mutations. We first used the previous parameters (e.g., Chromosome = 1 Mb, HGT = 20 kb, and population size = 10,000) and tested three sets of experiments with a Gamma-distributed dataset, where the mean value was set to 1. The scale parameters for these experiments were set to $1 \times 10^2$, $1 \times 10^4$, $1 \times 10^5$, and $1 \times 10^6$. The timing of selection and the fitness function were the same as those used in the third simulation scenario.

To better approximate the real KP genome, we adjusted the chromosome length to 5 Mb, set the mutation rate to $10^{-6}$, and increased the number of HGT events to 5, with each event transferring 20 kb, resulting in a total of 100 kb HGT per generation between each bacterium and others in the population. Each locus on the chromosome was assigned a selection coefficient drawn from a Gamma distribution, which included 5 million values, one for each locus on the chromosome, regardless of whether a mutation occurred at that locus. The scale for the Gamma distribution was set to $5 \times 10^6$ for this simulation. Similarly, the mean value of the Gamma distribution was set to 1. Fitness was determined by a normalized power function of the deviation from the mean score ($fitness = 2.0 + 0.5 \times (|score - \mu|^{10}) / (max(|score - \mu|^{10}) + 10^{-6})$, where $\mu$ is the mean score across all individuals). Based on the trait value–derived fitness (ranging from 2.0 to 2.5), individuals with more extreme trait values had higher fitness.

## Supporting information

**S1 Table. Enrichment of KP1_0338 high and low gene types among different host sources based on Fisher's exact test.** For each host, a one-vs-rest Fisher's exact test was performed to evaluate whether the gene type distribution differed significantly from that of all other sources combined.
(XLSX)

**S2 Table. Metadata of the non-redundant KP genome dataset.** Metadata for the 369 non-redundant KP genomes selected based on pairwise core SNP distances >20 kb. The table includes the Genome Context File ID (CGF ID), host type, isolation source, and geographical origin for each genome.
(XLSX)

**S3 Table. Metadata of the non-redundant VP genome dataset.** Metadata for the 667 non-redundant VP genomes selected based on pairwise core SNP distances >50 kb. The table includes genome identifiers and accession information, data source and BioSample records, isolate identifiers, year of isolation, geographic origin (continent, country, and country-level detail), isolation type, sequence type (ST), and population assignment.
(XLSX)

**S4 Table. Metadata of the multi-host KP genome dataset used for host specificity analyses.** Genomes were selected based on assembly size, contiguity, GC content, and annotation quality across MLST, rMLST, and SCGMLST schemes, and species classification was refined using Kleborate.
(XLSX)

**S1 Fig. Genomic characteristics of the non-redundant KP dataset. (A)** SNP distance matrix illustrating the genetic distances between strains in the non-redundant dataset, with strains ordered based on their PC1 values. The minimum SNP distance observed is 20,000 SNPs. **(B)** fineSTRUCTURE analysis of the non-redundant KP dataset, highlighting population structure and clustering patterns. The "strain" bar on the left represents the PC1 values of the respective strains. The data underlying this Figure can be found in https://zenodo.org/records/18520201.
(TIF)

**S2 Fig. Phylogenetic trees of the non-redundant and full KP datasets. (A)** Maximum-likelihood tree of the non-redundant *K. pneumoniae* dataset generated with IQ-TREE. **(B)** Maximum-likelihood tree of the complete dataset generated with FastTree. The data underlying this Figure can be found in https://zenodo.org/records/18520201.
(TIF)

**S3 Fig. Illustration of the half-sample splitting based on the phylogenetic backbone. (A)** Phylogenetic tree of the non-redundant KP dataset divided into two subsets based on the backbone. **(B)** PCA of the non-redundant KP dataset colored according to two sub-groups. The data underlying this Figure can be found in https://zenodo.org/records/18520201.
(TIF)

**S4 Fig. Half-samples test for the clonal simulation. (A)** SNP distance matrix for the clonal simulation. The minimum SNP distance observed is 2,000 SNPs. Half-strain matching: **(B)** Phylogenetic tree of the non-redundant clonal simulation divided into 2 subsets. **(C)** PCA of the non-redundant clonal simulation data. **(D)** Correlation of PC1 from group 2 projected onto group 1 results. **(E)** Correlation of PC1 from group 1 projected onto group 2 results. The data underlying this Figure can be found in https://zenodo.org/records/18520201.
(TIF)

**S5 Fig. Half-genome and Partial-variants tests for the clonal simulation. (F)** Division of the genome into two fragments (Fragment 1 and Fragment 2). **(G)** Correlation of PC1 between the two fragments. **(H)** Histogram of SNP loadings

on PC1, dividing SNPs into high- and low-loading groups. **(I)** Correlation of PC1 between high- and low-loading SNP groups. The data underlying this Figure can be found in https://zenodo.org/records/18520201.
(TIF)

**S6 Fig. Half-matching tests for VP datasets. (A)** fineSTRUCTURE analysis assigning populations for the non-redundant VP dataset. **(B)** Representation of VP Chromosome 1 and Chromosome 2, with proportions reflecting their actual sizes. Half-strain matching for the non-redundant VP dataset: **(C)** Correlation of PC1 from group 2 projected onto group 1 results; **(D)** Correlation of PC1 from group 1 projected onto group 2 results. Half-chromosome matching for the non-redundant VP dataset: **(E)** Correlation of PC1 between the two chromosomes. Half-matching for VppAsia dataset: **(F)** Correlation of PC1 from group 2 projected onto group 1 results; **(G)** Correlation of PC1 from group 1 projected onto group 2 results. Half-chromosome matching for VppAsia dataset: **(H)** Correlation of PC1 between the two chromosomes. The data underlying this Figure can be found in https://zenodo.org/records/18520201.
(TIF)

**S7 Fig. Zoomed-in views of four high-loading SNP peaks identified across the genome.** Each panel shows a zoomed-in genomic region corresponding to a high-loading SNP peak, in which multiple SNPs are clustered within a genomic interval of less than 10 kb. The data underlying this Figure can be found in https://zenodo.org/records/18520201.
(TIF)

**S8 Fig. Overview of block 1 from LD analysis with variant status heatmaps. (A)** SNPs in KP1_0337, KP1_0338, and associated accessory genes form a high LD block. **(B)** Variant (1/0) heatmap shows clear differentiation in gene presence/absence or SNP minor/major alleles across isolates. **(C)** Multi-species phylogenetic tree of the KP1_0338 gene. The annotation strip indicates the species of each strain. COG one-letter code descriptions: [N] Cell motility; [U] Intracellular trafficking, secretion, and vesicular transport. The data underlying this Figure can be found in https://zenodo.org/records/18520201.
(TIF)

**S9 Fig. Overview of block 2 from LD analysis with variant status heatmaps. (D)** High LD observed for SNPs in KP1_1632. **(E)** Variant (1/0) heatmap highlights distinct patterns of gene presence/absence or SNP minor/major alleles across isolates. COG one-letter code descriptions: [U] Intracellular trafficking, secretion, and vesicular transport. The data underlying this Figure can be found in https://zenodo.org/records/18520201.
(TIF)

**S10 Fig. Overview of block 4 from LD analysis with variant status heatmaps. (F)** High LD observed for variants in KP1_5387, forming a small, highly linked cluster. **(G)** Variant (1/0) heatmap shows distinct variation in gene presence/absence or SNP minor/major alleles across isolates. COG one-letter code descriptions: [T] Signal transduction mechanisms. The data underlying this Figure can be found in https://zenodo.org/records/18520201.
(TIF)

**S11 Fig. Overview of block 3 from LD analysis with variant status heatmaps. (H)** SNPs in KP1_RS09345, KP1_RS30360, KP1_1982, and accessory genes form tightly linked clusters. **(I)** Variant (1/0) heatmap reveals clear differentiation in gene presence/absence or SNP minor/major alleles across isolates. COG one-letter code descriptions: [C] Energy production and conversion; [G] Carbohydrate transport and metabolism; [K] Transcription; [K] Transcription; [M] Cell wall/membrane/envelope biogenesis; [O] Posttranslational modification, protein turnover, chaperones; [P] Inorganic ion transport and metabolism; [Q] Secondary metabolites biosynthesis, transport and catabolism; [S] Function unknown; [T] Signal transduction mechanisms. The data underlying this Figure can be found in https://zenodo.org/records/18520201.
(TIF)

**S12 Fig. Phylogenetic tree of KP1_0338 within KP and host origin profiles. (A)** Phylogenetic tree of the KP1_0338 gene from the multi-host KP dataset, with annotation strips indicating host origin and the KP1_0338 high/low classifications inferred from clustering. The canonical reference strains MGH78578 (low) and NTUH-K2044 (high) are labeled for reference. **(B)** Projected the multi-host dataset onto the non-redundant dataset. Each bar represents the mean PC1 value for isolates from a given host, with error bars indicating the standard deviation. The data underlying this Figure can be found in https://zenodo.org/records/18520201.
(TIF)

**S13 Fig. Clonal haploid bacteria model with varying HGT lengths and a population size of 2000.** Phylogenetic trees, PCA plots, and scree plots for a population of 2000 individuals under different HGT lengths. Panels show results for: **(A, B)** 100 bp, **(C, D)** 1 kb, **(E, F)** 10 kb, **(G, H)** 100 kb. The data underlying this Figure can be found in https://zenodo.org/records/18520201.
(TIF)

**S14 Fig. LD decay under different HGT lengths in clonal haploid bacteria models.** Comparison of LD decay between different HGT lengths (100, 1, 10, and 100 kb) and the non-redundant KP dataset. **(A)** Population size of 1,000. **(B)** Population size of 2,000. **(C)** Population size of 10,000. The x-axis represents the SNP distance (log scale), and the y-axis represents the LD value ($R^2$) on a log scale. The data underlying this Figure can be found in https://zenodo.org/records/18520201.
(TIF)

**S15 Fig. Linear stepping-stone model with varying numbers of migrants and a population size of 2000.** Phylogenetic trees, PCA plots, and scree plots for a model with HGT of 10 kb. The population consists of 10 subpopulations (200 individuals each). Panels show results for different numbers of migrants exchanged between neighbor subpopulations: **(A, B)** 1 migrant, **(C, D)** 10 migrants, **(E, F)** 20 migrants, **(G, H)** 50 migrants. The data underlying this Figure can be found in https://zenodo.org/records/18520201.
(TIF)

**S16 Fig. LD decay profiles are compared between linear stepping-stone models with varying numbers of migrants (1, 10, 20, 50) and the non-redundant KP dataset.** Panels illustrate results for different total population sizes: **(A)** 2000 individuals, **(B)** 10,000 individuals, **(C)** 100,000 individuals. Each model includes HGT of 10 kb. The x-axis represents the SNP distance (log scale), and the y-axis represents the LD value ($R^2$) on a log scale. The data underlying this Figure can be found in https://zenodo.org/records/18520201.
(TIF)

**S17 Fig. Heterogeneity models.** Recombination-rate heterogeneity model: **(A)** Maximum-likelihood phylogenetic tree, **(B)** PCA of the simulated genomes, **(C)** Circular Manhattan plot showing SNP-wise squared PC1 loadings. Regions with elevated recombination rates appear as depressions in loading values. Mutation-rate heterogeneity model: **(D)** Maximum-likelihood phylogenetic tree, **(E)** PCA of the simulated genomes, **(F)** Circular Manhattan plot of squared PC1 loadings. The data underlying this Figure can be found in https://zenodo.org/records/18520201.
(TIF)

## Author contributions

**Conceptualization:** Daniel Falush.

**Formal analysis:** Siqi Liu.

**Methodology:** Sarah L. Svensson.

**Writing – original draft:** Siqi Liu, Sarah L. Svensson, Daniel Falush.

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
