## [Editor Report · Decision Letter 0]

14 Aug 2025

Dear Daniel,

Thank you for submitting your manuscript entitled "A putative bacterial ecocline in Klebsiella pneumoniae" for consideration as a Research Article by PLOS Biology.

Your manuscript has now been evaluated by the PLOS Biology editorial staff, as well as by an academic editor with relevant expertise, and I'm writing to let you know that we would like to send your submission out for external peer review.

Once your full submission is complete, your paper will undergo a series of checks in preparation for peer review. After your manuscript has passed the checks it will be sent out for review. To provide the metadata for your submission, please Login to Editorial Manager (https://www.editorialmanager.com/pbiology) within two working days, i.e. by Aug 18 2025 11:59PM.

Kind regards,

Roli

Roland Roberts, PhD

Senior Editor

PLOS Biology

rroberts@plos.org

---

## [Decision Letter · Decision Letter 1]

7 Oct 2025

Dear Daniel,

Thank you for your patience while your manuscript "A putative bacterial ecocline in Klebsiella pneumoniae" was peer-reviewed at PLOS Biology. It has now been evaluated by the PLOS Biology editors, an Academic Editor with relevant expertise, and by four independent reviewers.

You'll see that reviewer #1 is overall positive, but thinks that the limitations of the study should be spelled out more explicitly, and the paper should be made more accessible to the broader reader. Reviewer #2 also votes for improved accessibility, and suggests a minor analysis. Reviewer #3 is broadly positive, but thinks that you overclaim, suggesting several potential alternative suggestions (heterogeneous drift, changes in recombination rate, historical contingency…); s/he suggests a number of additional analyses and requests some clarifications. Reviewer #4, like reviewer #3, is intrigued, but not fully convinced by the strength of support. Again, more explicit consideration of alternative explanations is warranted, and there are a few extra analyses among the requests.

In light of the reviews, which you will find at the end of this email, we would like to invite you to revise the work to thoroughly address the reviewers' reports.

Given the extent of revision needed, we cannot make a decision about publication until we have seen the revised manuscript and your response to the reviewers' comments. Your revised manuscript is likely to be sent for further evaluation by all or a subset of the reviewers.

**IMPORTANT - SUBMITTING YOUR REVISION**

*Re-submission Checklist*

*Published Peer Review*

*PLOS Data Policy*

Sincerely,

Roli

Roland Roberts, PhD

Senior Editor

PLOS Biology

rroberts@plos.org

REVIEWERS' COMMENTS:

Reviewer #1:

This is an interesting and innovative study that uses the population structure and genomic diversity of Klebsiella pneumoniae (KP) to infer the action of selective pressure on a quantitative trait, where extreme values appear more advantageous than intermediate ones. The study relies entirely on genomic analyses (comparisons with simulations and across species), which limits its ability to identify the underlying trait under selection, leaving this open for future work. The manuscript is generally clear, and the arguments are well supported by the comparative analyses. However, it may be difficult for readers outside the field to follow, and in some cases the conclusions slightly overstate what can be inferred from the analyses. I therefore suggest that the authors moderate some statements, and add brief additional discussion or analysis of possible confounders and alternative explanations.

The study puts forward a compelling explanation for the structure of the KP phylogeny, supported by a logical set of comparative analyses (simulations and cross-species comparisons). While the authors highlight some genes strongly associated with this cline, they do not identify the trait(s) driving the structure. This is somewhat disappointing, as experimental assays could have helped explore links between quantitative traits and the observed genetic variation. That said, there is no guarantee that quantitative traits measurable in laboratory assays would capture the true driver of this subtle structuring of the global KP population. The study therefore provides evidence consistent with the existence of a quantitative trait and points to some possible functional candidates.

The argument that selection on a quantitative trait could explain this structure is convincing, though alternative explanations have not been fully excluded. This is inherently difficult, since simulations only capture simplified scenarios, and the dataset is affected by inevitable sampling biases. The authors aim to explain the 'backbone' structure, the overrepresentation of samples at its extremes, and the uneven genomic associations with the backbone. Taken together, these features point towards diversifying selection on a quantitative trait as a likely driver. Nonetheless, given the complexity of bacterial genome evolution and population structure, unknown aspects of evolutionary history and ecology, and the potential for sampling biases, some caution is warranted.

The authors explore possible links between metadata and population structure (geography, virulence, resistance), but they might also consider other confounders such as sampling date, host source, genome size, and base composition. While they touch on host and environmental sources, and their connection to geography, this could be discussed earlier as a possible driver of the patterns observed. They might also reflect on potential (including unrecognised) sources of sampling bias. For example, if all isolates were cultured before sequencing, the growth conditions themselves may have imposed selective pressures.

It would also be helpful if the discussion acknowledged how more complex scenarios could generate the observed patterns. For instance, while geographic isolation alone may explain the backbone but not other features, could combining isolation with restricted recombination (e.g. isolation by distance) do so? Could the apparent lack of clear geographic structure reflect reduced geographic differentiation in recent years due to greater movement of humans and livestock? In addition, the simulations do not include the accumulation of variation through interspecies recombination (which the authors observe in their block 1). Could the combination of geographic structure and constraints on recombination explain the observed patterns? While it would be unreasonable to expect the authors to model all such scenarios, it would strengthen the manuscript to either explain why these more complex alternatives are unlikely explanation of the observed patterns, or acknowledge that they remain possible.

Although clearly written overall, the manuscript is likely to be challenging for non-specialists in population genetics or genomics. Adding more explanation of the reasoning behind the analyses would make the work more accessible, particularly for readers from other disciplines who might build on these findings through experimental or observational work (as the authors suggest through a 'reverse ecology' approach).

Finally, there are a few places where the strength of the conclusions is overstated, particularly where no clear threshold distinguishes presence or absence of a given pattern. For example, in lines 124-125 the authors claim that they can 'falsify the simplest neutral model' based on their results. This goes beyond what can be concluded: no formal test is presented, and absence of evidence (of similar structure) does not amount to evidence of absence.

Reviewer #2:

This study addresses the population genetic structure of Klebsiella pneumoniae (KP), a species known for global dispersal and high recombination rates. Phylogenies generated here reveal a backbone-like structure, corresponding to a major axis of variation (PC1) that explains 16.8% of genetic diversity. The authors propose that this reflects a "bacterial ecocline" shaped by diversifying selection on a quantitative trait rather than solely demographic processes. Using simulations they demonstrate which parameters can reproduce observed KP genetic patterns. The association of PC1 with loci including kpa fimbriae genes, suggests modulation of adhesive properties could be driving the evolution. This hypothesis was not empirically proven, but that seems outside the scope of the study.

Major comments:

1. As a reviewer outside of evolutionary biology, I had a difficult time understanding some of the language used. To make this article more broadly accessible and, thus, more impactful, I suggest providing more explanation of the jargony terms. Some examples include " PC1 loadings", "absolute PC1 loading values", "LD decay curve", "trait value density", "Gamma distribution", etc.

2. It seems like the 4 SNP sites are all distributed within 2.5 Mb of each other. Where do the 4 SNP enrichment sites (Fig 4A) fall on the DNA schematic in 2C? Are the 4 sites split evenly between left-hand and right-hand groups? If they were not split evenly between left and right side, I wonder if Fig 2 is repeated with the 4 SNP sites weighted on the left-side and no SNP sites fall on the right side, if the ecocline would be lost on the right side. This could strengthen the conclusion that the 4 SNP sites are driving the observed backbone.

Minor comments:

3. Generally figure axis labels and figure-embedded text are small and difficult to read. Also, it is difficult to discriminate the blue trace of 'real data' vs the blue traces of some simulated data (throughout manuscript). Perhaps switch 'real data' to black, green, or red traces so it is clearly distinguished from the simulated data. Or use a heavier weight line.

4. Fig 3 section references "Molassodon" ecospecies in the text, but does not identify them in Fig 3. Is Eg1a Molassodon ecospecies. If not, what is Eg1a vs non-Eg1a?

5. Line 152 typo - "dataset of," the dataset is not defined.

6. I believe all genome loci are referred to using NTUH-K2044 locus tags. It would be helpful to explicitly state that early in the Fig 4 data section. Perhaps report that in Fig 4 and Fig S6 legends.

7. Cross-check the results text and Supplementary Fig 7 legend. It appears that panel orders were switched and some Fig S7 panels are not identified accurately.

8. Line 232 - I think 'solid (a) and dashed (b)' should be switched to 'dashed (a) and solid (b)'

9. Would it be possible to write COG letters in Fig 5B and 5C? This could help readers link COG letters embedded in Fig S7. Alternatively, could the function of the COG letters in Fig S7 be provided in the legends?

10. Fig S8 - States that panel A show KP1_0338 high or low, but MGH vs NTUH is just identified. It would help reader comprehension if the KP1_0338 high and low strains are identified alongside their name in the figure panel.

Reviewer #3:

This paper explores why there is a "backbone" -- a connected path of relatively long internal branches -- in phylogenies of K. pneumo.

The authors argue that such a pattern is not observed in simple neutral models and is likely due to an "ecocline": a quantitative trait under diversifying selection.

I really appreciate some aspects of this: the big-picture view and interest in a fundamental evolutionary question; the use of the tree shape; the use of simulations to help explore what-if scenarios (rather than, say, simply assuming that a model such as one with diversifying selection must have given rise to the data, then estimating a parameter in that model, then believing that estimate). However, there are some major limitations and a few instances where I think the authors have over-claimed.

The scientific argument is that diversifying selection on a quantitative trait can produce an observed pattern, and maybe that it is plausible (though the trait is yet to be identified). It is hard to argue that it is the only thing that could plausibly produce the pattern, or, to even characterize what else might be able to produce the same pattern. What happens in the model if there is no selection, just drift, but with a heterogenous distribution

of drift rates across the genome? What about changes in the recombination rate over time, with deep branches not having much, and then more movement? would historical contingency create an apparent backbone?

In order of appearance in the manuscript (not importance):

How much phylogenetic support is there for the deep branches that define the "backbone"?

Phylo reconstruction algorithms are known to produce "ladder shapes" where there

isn't much support for a different branching pattern; there is long branch attraction, etc. The fact that the same backbone appears in different trees is reasonable evidence but you could try raxml or iqtree especially where you have only 100s of tips, not 1000s. Even for 1000s these are very feasible.

Phylogenetically, the backbone is a connected path consisting of relatively long internal branches, meaning that they are longer by quite a bit than other internal branches. And it's the only such path, and only branches in a certain length range are in it (pretty much). Does this define the backbone?

They are shorter than the pendant branches and consistently longer than all other

internal branches. You could quantify this property rather than relying on the layout to define it (since tree layout algorithms can be very misleading). That way, you could also see to what extent it appears in simulations, without relying on visualizing trees one by one to see whether a backbone is emerging. You could define a backbone in terms of the number of (connected) longer internal branches, their depth in the tree, and their relative length (compared to the other internal branches and compared to pendant branches).

Line 104 - re left/right split and clonal structure: no, the right and left parts of the tree aren't the clonal structure that the tree captures (such as it is), because there is no split in the tree corresponding to left/right. In fact the PCA *is* connected to the clonal structure, as you can see in figure 1 with the colours. But because the clonal structure is weak, the pca axes capture little of the variation and aren't too connected to the tree. Suggest removing the artificial left-right split. It's also very dependent on the layout algorithm.

p5 line 125 this seems like an overclaim - you can't falsify a whole model class based on a few simulations and a PCA.. Fitting the model parameters might allow you to capture the observed patterns (which anyway you haven't quantified).

p7 line 163 onwards I was losing the story a bit. Why are these results about genes and their loading on PCA1 and their LD here?

Why not instead look at the splits of the taxa that are indicated by the edges defining the backbone? These would be the most direct questions to suggest what the trait might be , if there is one, because whatever the gene/ SNP differences are along the backbone describes the evolution that defines the backbone. If there is strong phylogenetic support for those edges ,there is presumably also support for the changes that those edges define.

L188 what does "encoding the usher" mean?

page 10 -- i don't know where the approximately causal language comes from. PCA vectors are defined by the data. They don't have "influence". They aren't, in this case, even population structure. Again (see previous comment) I wondered about the long branches in the backbone; would it not be more interesting to look at which mutations are different from top to bottom of this backbone? How similar would this be to the PCA? judging from fig 1a there would be some similarity, but they are not the same.

Line 247 re virulence, AMR, hosts, etc: did you test for phylogenetic signal ("clustering")? can we see the trees with the colours corresponding to virulence, amr, geography, host, etc?

It wouldn't surprise me if virulence (in humans) doesn't matter; why would it (except for sampling artifacts, several of which you go on to describe clearly). Anyway, visually not observing clustering (and not even showing it) isn't strong evidence; there can be phylogenetic signal that isn't immediately apparent visually.

page 12 -actually supplementary fig 9g is getting there! i think I see a backbone.

what happens as you change other parameters, like N?

More broadly, why are the parameters you choose for the simulations very likely to be the best ones for trying to find the effect you are trying to find? (Likely it is hard to know a priori, but this means that trying a few out and not seeing the phenomenon you are interested in is quite weak evidence).

Without a quantitative description of the phenomenon you are interested in (see above comment) you can't assess to what extent it may be emerging in simulations.

What about: heritable high substitution rate that is then lost, but whose effects remain, deep in the tree?

A lot is made of the visual appearance of the tree. Even if you want to stick to visual comparisons, you could look at what the tree looks like in a rooted view? with the root at one end of the backbone,? with the root in the middle? etc. Rooted trees can suggest different interpretations, which might be interesting.

Overall I think it's interesting. but having done a lot of modelling, i would be very cautious in

simulating something, seeing it not match the data, and then concluding that the model *cannot* match the data. There is usually a **lot** of work between "this didn't look like the data when I simulated with my first guess at the parameters" and "this model *cannot* generate a pattern like the one seen in data".

Line 596: iterative popPUNK selection: what does this mean? Please say more about what an isolate needs to do to get into the data. What sampling (tended to) result in

inclusion in the poppunk or kleborate samples?

I don't work with this organism but if you have core SNPs in 99% of the isolates doesn't that mean you have a poor choice of reference?

Line 613 fix grammar.

What's R2 in haploview? some explanation for those not familiar with it would help.

line 756 Isolation → isolate?

With recombination rates - line 835 - what is the interpretation of "generation"? presumably even KP doesn't get recombination opportunities, never mind successes, on every cell division? what happens to the simulated data and its ecocline (or lack of) as you change simulation parameters?

Reviewer #4:

In this manuscript, Liu et al., first analyse the population structure of Klebsiella pneumoniae (KP), describing a phylogenetic structure with lineages radiating from a backbone. They then propose a model in which diversifying selection on a quantitative trait reproduces this structure. The authors hypothesise the backbone therefore reflects a bacterial ecocline.

I found this work very interesting and the insights valuable - particularly in introducing quantitative traits in our thinking about bacterial evolution. The overall approach is convincing, though I do not have the right expertise for an in-depth technical review of the genomic analyses. Given its complexity, the manuscript is clearly written. I did not find the evidence for the ecocline model entirely convincing (in particular because of how alternative explanations were treated - see below), however I do not think this is a problem: the authors are very clear this model is presented as a hypothesis.

In my view, this work is a good fit for Plos Biology, but I do have a few points that I think are very important to address and some minor suggestions.

Essential points:

1. I was not able to find any details on the genomic data (other than most of them being from humans, line 250), which makes interpreting the genomic analyses more challenging. At a minimum, it would be helpful to have the total number of isolates, and the distribution of isolates from different host types and different geographical locations. If the information is available, it would also be very helpful to know whether human and environmental isolates are hospital-associated or from the community.

2. For all of the modelling results, it was unclear to me whether the 10k generations represent an equilibrium state. Are the results (e.g. shape of the phylogenies, presence of intermediate trait values) robust to running the simulations for longer? In particular, it was not clear from the results as presented here whether the model predicts a stable phylogenetic backbone, or whether this is a temporary feature and simulations eventually converge to either two distinct populations or a population with no structure.

3. I did not entirely follow the reasoning for dismissing heterogeneity in recombination rates between strains / ecological demes as an explanation for the phylogenetic backbone.

The authors exclude heterogeneity in recombination rates in part because this reduces the effectiveness of recombination" and therefore likely makes it difficult to recapitulate the low levels of LD in KP at large genetic distances (line 296). I do not follow the logic here, one could also frame these models as increasing the effectiveness of recombination in closely related strains relative to distantly related species. More generally, the authors parameterise recombination based on fitting the LD decay curve - it would seem surprising if including heterogeneity in recombination rate (which increases the dimensionality of the model), would make it difficult to fit this curve. I understand the authors also dismiss this explanation based on previous results, however, the argument would be considerably stronger if this alternative model were explicitly explored.

I understood the reasoning for why the geographical model is an unlikely explanation for the data, as there is no evidence of geographical structuring, but not why this structuring could not arise from ecological demes (line 329). If the analysed genomes are primarily from humans, the demes would have to exist within the human population (e.g. age structure or hospital vs community), so the argument about acquisition of KP by humans from other sources does not seem relevant.

Overall, I think the manuscript would benefit from a clearer discussion of the ecocline hypothesis in relation to alternative hypotheses and ideally from additional simulations of alternative models, particularly relating to heterogeneity in recombination rate.

Minor points:

Line 243: "our analysis suggests that they are not the primary determinants of its overall population structure, which is not unexpected given that most lineages only cause human disease sporadically." This is not relevant in the context of antibiotic resistance, as KP is also exposed to antibiotics when carried asymptomatically ("bystander selection").

Line 847: In the model, the average number of offspring is >1. Is the population therefore growing, or is there an additional step to maintain constant population size?

Supplementary table 1: I thought most of the isolates were from humans, but this is missing in the table.

Some of the figures are quite difficult to read - e.g. very small font size on Figs 2 and 4, difficulty to distinguish colours and lines on Fig 7j.

Thank you for making the code available. I suggest adding a readme to the github directory to help others navigate the code.

---

## [Editor Report · Decision Letter 2]

30 Jan 2026

Dear Daniel,

Thank you for your patience while we considered your revised manuscript "A putative bacterial ecocline in Klebsiella pneumoniae" for publication as a Research Article at PLOS Biology. This revised version of your manuscript has been evaluated by the PLOS Biology editors and the Academic Editor.

Based on our Academic Editor's assessment of your revision, we are likely to accept this manuscript for publication, provided you satisfactorily address the following data and other policy-related requests.

IMPORTANT - please attend to the following:

a) Please could you make your Title more explicit; we would suggest something like "A bacterial ecocline in Klebsiella pneumoniae may explain its backboned phylogeny" (I'm not sure whether "backboned" is a word, but I think it may intrigue readers in the same way that your poster at SMBE did me)?

b) Please address my Data Policy requests below; specifically, we need you to supply the numerical values underlying Figs 1AB, 2ABCDEFGH, 3ABCDEFGHIJKL, 4AB, 5ABC, 6ABCEFG, 7ABCDEFGHIJK, 8ABCDEFGHIJK, 9ABCDEFGH, S1AB, S2AB, S3AB, S4ABCDEGHI, S5ACDEFGH, S6, S7ABCDEFGHI, S8AB, S9ABCDEFGH, S10ABC, S11ABCDEFGH, S12ABC, S13ABCDEF (including treefiles or equivalent), either as a supplementary data file or as a permanent DOI’d deposition.

c) Please cite the location of the data clearly in all relevant main and supplementary Figure legends, e.g. “The data underlying this Figure can be found in https://zenodo.org/records/XXXXXXXX

d) Please make any custom code available, either as a supplementary file or as part of your data deposition.

e) Please include the URLs of your funders in the Financial Disclosure statement.

We expect to receive your revised manuscript within two weeks.

*Published Peer Review History*

*Press*

Sincerely,

Roli

Roland Roberts, PhD

Senior Editor

rroberts@plos.org

PLOS Biology

DATA POLICY:

Regardless of the method selected, please ensure that you provide the individual numerical values that underlie the summary data displayed in the following figure panels as they are essential for readers to assess your analysis and to reproduce it: Figs 1AB, 2ABCDEFGH, 3ABCDEFGHIJKL, 4AB, 5ABC, 6ABCEFG, 7ABCDEFGHIJK, 8ABCDEFGHIJK, 9ABCDEFGH, S1AB, S2AB, S3AB, S4ABCDEGHI, S5ACDEFGH, S6, S7ABCDEFGHI, S8AB, S9ABCDEFGH, S10ABC, S11ABCDEFGH, S12ABC, S13ABCDEF (including treefiles or equivalent). NOTE: the numerical data provided should include all replicates AND the way in which the plotted mean and errors were derived (it should not present only the mean/average values).

CODE POLICY

Per journal policy, if you have generated any custom code during the course of this investigation, please make it available without restrictions. Please ensure that the code is sufficiently well documented and reusable, and that your Data Statement in the Editorial Manager submission system accurately describes where your code can be found. More information on our Code Policy, what and how to share can be found here: https://journals.plos.org/plosbiology/s/code-availability

DATA NOT SHOWN?

---

## [Editor Report · Decision Letter 3]

11 Feb 2026

Dear Daniel,

Thank you for the submission of your revised Research Article "A bacterial ecocline in Klebsiella pneumoniae may explain its backboned phylogeny" for publication in PLOS Biology. On behalf of my colleagues and the Academic Editor, Kathryn Holt, I'm pleased to say that we can in principle accept your manuscript for publication, provided you address any remaining formatting and reporting issues. These will be detailed in an email you should receive within 2-3 business days from our colleagues in the journal operations team; no action is required from you until then. Please note that we will not be able to formally accept your manuscript and schedule it for publication until you have completed any requested changes.

Sincerely,

Roli

Senior Editor

PLOS Biology

rroberts@plos.org